*J Physiol* 603.7 (2025) pp 2139–2156                                                  2139

# Cardiac neuromodulation with acute intermittent hypoxia in rats with spinal cord injury

Mehdi Ahmadian[1,2,3,4], Erin Erskine[1,2,3] and Christopher R. West[1,2,3]

[1] *International Collaboration on Repair Discoveries, University of British Columbia, Vancouver, BC, Canada*
[2] *Centre for Chronic Disease Prevention and Management, University of British Columbia, Kelowna, BC, Canada*
[3] *Department of Cellular and Physiological Sciences, Faculty of Medicine, University of British Columbia, Vancouver, BC, Canada*
[4] *Breathing Research and Therapeutics Centre, Department of Physical Therapy and McKnight Brain Institute, University of Florida, Gainesville, FL, USA*

Handling Editors: Harold Schultz & Diana Martinez

The peer review history is available in the Supporting Information section of this article (https://doi.org/10.1113/JP287676#support-information-section).

**Abstract figure legend** Using a sequential pharmacological blockade design, we investigated the roles of both branches of the autonomic nervous system in cardiac dysfunction following an experimental spinal cord injury (SCI) in rats. Cardiac dysfunction was found to be primarily driven by impaired sympathetic control post-SCI. Exposing SCI rats to acute intermittent hypoxia (AIH), unlike time control (TC), neuromodulated the heart and enhanced cardiac function, evidenced by augmented left-ventricular pressure-generating capacity ($dP/dt_{max}$).

**Abstract** It is well recognized that the interruption in bulbospinal sympathetic projections is the main cause of cardiovascular instability in individuals and experimental animals with spinal cord injury (SCI). Whether interrupted bulbospinal sympathetic projections contribute to cardiac dysfunction directly (i.e. input to the heart) or indirectly (i.e. vascular influences that alter loading conditions on the heart) post-SCI remains unknown, as does the potential effect of SCI-induced alterations in parasympathetic control on heart function. We employed a sequential pharmacological blockade approach to bridge this knowledge gap and additionally examined whether acute intermittent hypoxia (AIH) is capable of neuromodulating the heart post-SCI. In two experiments, rats were given T3 contusion SCI and survived for 2 weeks. At 2 weeks post-SCI, rats were instrumented with left ventricular and arterial catheters to assess cardiovascular function in response to either a sequential pharmacological blockade targeting different sites of the autonomic neuraxis (experiment 1) or AIH (experiment 2). The findings from experiment 1 revealed that impaired direct sympathetic transmission to the heart underlies the majority of the SCI-induced reduction in heart function post-SCI. The findings from experiment 2 revealed that a single-session of AIH increased left ventricular pressure generation and arterial blood pressure immediately and up to 90 min post-AIH. Together, our findings demonstrate that disrupted bulbospinal sympathetic pathways contribute directly to the SCI-induced impairment in left ventricular function. We also show that a single session of AIH is capable of neuromodulating the heart post-SCI.

(Received 11 September 2024; accepted after revision 26 February 2025; first published online 22 March 2025)

**Corresponding author** C. R. West: Centre for Chronic Disease Prevention and Management, Faculty of Medicine, Reichwald Health Sciences Centre, Kelowna, BC, Canada, V1V 1V7. Email: chris.west@ubc.ca

## Key points

- The loss of sympathetic transmission to the heart is the main cause of reduced cardiac function in a rodent model of spinal cord injury (SCI).
- Parasympathetic control remains unaltered post-SCI and does not contribute to reduced cardiac function post-SCI.
- Acute intermittent hypoxia neuromodulates the heart and increases left ventricular pressure generating capacity.

## Introduction

Humans (and animals) with spinal cord injury (SCI) suffer from numerous physiological sequelae, including respiratory, metabolic and endocrine, cardiovascular, gastrointestinal and urogenital dysfunction (Bauman & Spungen, 2000; Cardozo, 2007; Ebert, 2012; Hagen, 2015). Of these, cardiovascular complications are now one of the leading causes of morbidity and mortality in individuals with SCI (Cragg et al., 2013; DeVivo et al., 2022; Peterson et al., 2021). The odds of stroke and heart disease are roughly three times higher in individuals with SCI than in able-bodied individuals (Cragg et al., 2013). Furthermore, in addition to causing conditions such as autonomic dysreflexia (AD) and orthostatic hypotension associated with disrupted autonomic outflow post-SCI (Eldahan &

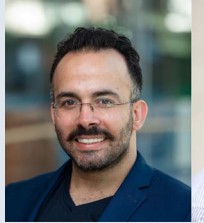 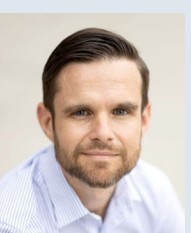

**Mehdi Ahmadian** (left) is a Gator Neuroscholar and postdoctoral associate at the McKnight Brain Institute, University of Florida. This published work originates from his PhD research conducted under the mentorship of **Christopher R. West** (right picture) in the Translational Integrative Physiology Laboratory at the University of British Columbia, Canada. One of the main focuses of Dr West's lab is cardiovascular autonomic function and the effects of spinal cord injury, emphasizing the mechanisms of dysfunction and the development of therapeutic strategies. Dr Ahmadian's research focuses on neuroscience, cardiorespiratory physiology and translational medicine. He explores advanced therapeutic approaches, including acute intermittent hypoxia and physiological interventions, aiming to mitigate dysfunction in neurological and systemic conditions. His goal is to translate foundational physiological insights into practical treatments, addressing challenges in neurodegenerative diseases, spinal cord injury and disorders affecting autonomic and cardiorespiratory health.

Rabchevsky, 2018; Krassioukov et al., 2014), high-level SCIs also cause an immediate and significant reduction in cardiac function, followed by subsequent changes in cardiac morphology in the chronic setting (Fossey et al., 2022; Poormasjedi-Meibod et al., 2019; Williams et al., 2019). Indeed, chronic high-thoracic/cervical SCI is associated with reduced contractile function (DeVeau et al., 2017; Squair et al., 2018; Williams et al., 2020), a down- and right-shifted starling curve (West et al., 2014, 2016), and a reduction in left ventricular (LV) chamber size (Kessler et al., 1986; Williams et al., 2019).

We have previously documented that high-thoracic SCI causes a rapid and sustained reduction in cardiac function that is associated with interrupted bulbo-spinal sympathetic control in both animal models and humans with chronic SCI (Fossey et al., 2022). However, whether interrupted bulbospinal sympathetic control contributes directly (i.e. altered input to the myocardium) or indirectly (i.e. altered vascular loading conditions) to the reduction in heart function post-SCI remains unknown. Concurrent with disrupted sympathetic control, there occurs an unopposed parasympathetic control following high-thoracic SCI, resulting in severe autonomic imbalance (Henke et al., 2022). Moreover, there is experimental evidence of neuroplasticity within key parasympathetic nuclei post-SCI (Lujan et al., 2014). Yet the influence of altered vagal control of heart function, and in particular of LV function, post-SCI has rarely been investigated. Outside of SCI, there is evidence of muscarinic fibres innervating the LV and relatively recent work suggests a tonic vagal influence on cardiac contractility (Coote, 2013; Machhada et al., 2016). Thus, it is possible that impaired cardiac function post-SCI is caused by a combination of alterations in both direct and indirect sympathetic control as well as altered parasympathetic control.

Neuromodulatory interventions are gaining widespread traction within the field of SCI. One of the main neuromodulatory options currently employed is acute intermittent hypoxia (AIH; episodic exposure of humans/animals to brief periods of low inspired oxygen concentration). AIH has previously been shown to effectively improve key motor functions such as walking, hand/arm movement and breathing in humans and animals with SCI (Vose et al., 2022). These AIH-induced beneficial effects have been attributed to neural plasticity manifesting as long-term facilitation (LTF; a long-lasting increase in motor nerve activity post-AIH), which for respiratory neural plasticity (i.e. phrenic LTF) can manifest functionally as ventilatory LTF (a long-lasting increase in minute ventilation post-AIH) in humans and awake animals (Griffin et al., 2012; Mitchell & Johnson, 2003; Panza et al., 2023; Vermeulen et al., 2020). Recently, we, and others, have documented that AIH also evokes sympathetic neuroplasticity (i.e. sympathetic LTF) in

spinally injured animals, albeit using different injury models (contusion *vs.* hemisection) and AIH protocols ($10 \times 1$ min *vs.* $3 \times 5$ min) (Ahmadian et al., 2025; Perim et al., 2023). The presence of AIH-induced sympathetic LTF post-SCI raises the intriguing possibility that AIH may yield functional benefits by neuromodulating the heart post-SCI.

In the present study, using sequential pharmacological blockade of autonomic transmission to the heart and vasculature in a rodent model of high-thoracic SCI, we first examined whether heart dysfunction following SCI is a consequence of either weakened direct or indirect sympathetic control, altered parasympathetic control or a combination of both. We subsequently examined whether one session of AIH applied 2 weeks following injury neuromodulates the heart in our rodent model of SCI. To address these aims, we used a contusion SCI model (i.e. leaving some neural pathways intact post-injury) to better mimic the clinical scenario of SCI ($\sim$70% incomplete) (National Spinal Cord Injury Statistical Centre, 2023). It was hypothesized that (1) altered (impaired) direct and indirect sympathetic control, not parasympathetic control, underlies impaired heart function post-SCI and (2) one session of AIH delivery neuromodulate the heart post-SCI.

## Methods

### Ethical approval

The study was conducted between January and September 2022. All experimental protocols and procedures in the present work were conducted in accordance with Canadian Council on Animal Care policies, with ethical approval obtained from the University of British Columbia Animal Care Committee (A18-0344). Animals were acclimated for at least 7 days prior to experimentation and group-housed in temperature-controlled rooms under 12:12 h light/dark photocycles and social/physical enrichments. Water and food were provided *ad libitum* during all studies. The authors confirm that they understand the ethical principles under which the journal operates and that their work conforms to the principles and regulations described in the Editorial by Grundy (2015).

### Experimental design

**Experiment 1.** Eighteen adult (10–12 weeks) male Wistar rats (300–350 g; Charles River Laboratories, Wilmington, MA, USA) were instrumented with LV and arterial catheters to allow for assessing cardiovascular function (Naive and SCI: $n = 9$ each). Rats subsequently underwent a sequential pharmacological

manipulation to examine whether, in addition to the well known sympathetic impact, the parasympathetic arm of the autonomic nervous system and reduced vascular influences on cardiac function also contribute to cardiac dysfunction post-injury. Sequential pharmacological blockade included a cardio-selective beta-1 adrenergic receptor antagonist (esmolol; 300 μg kg$^{-1}$ min$^{-1}$) to block sympathetic transmission to the heart, a muscarinic receptor antagonist (atropine methyl bromide; 4 mg kg$^{-1}$) to block parasympathetic transmission to the heart, esmolol plus atropine to fully block autonomic transmission to the heart (i.e. double blockade), and a cocktail of esmolol, atropine and hexamethonium bromide (30 mg kg$^{-1}$, a ganglionic blocker) to completely block autonomic transmission to the heart and vasculature. The latter enables the dissection of any additional vascular influences on cardiac function post-SCI. The order of drug administration was kept as described above as a result of the half-life of the drugs used. Although esmolol is considered an ultrashort-acting drug (with a half-life of 2 min, a time-to-peak effect of ∼6–10 min and a washout time of 9 min) (Wiest & Haney, 2012), the elimination half-life of atropine is ∼4 h (Adams et al., 1982). Cardiovascular function was assessed at baseline and during stable periods of each drug trial. Terminal assessment for SCI rats was performed at 2 weeks following SCI surgery.

**Experiment 2.** Nineteen adult (10–12 weeks) male Wistar rats (300–350 g; Charles River Laboratories) were given SCI and instrumented with cardiac and peripheral arterial catheters to assess cardiovascular responses to AIH and time control condition (TC, $n = 8$; AIH, $n = 11$). Cardiovascular function in response to AIH was assessed prior to, immediately when the final hypoxic was over (1 min post) and following (90 min post) AIH exposure at 2 weeks following SCI surgery.

### T3 contusion SCI surgery and care procedures

All procedures associated with surgically performing our T3 contusion injury and pre- and postoperative care have been described in detail previously (Wainman et al., 2021). Briefly, rats were initially anaesthetized using an inhalational anaesthetic (5% isoflurane chamber induction and maintenance on 1.5–2% isoflurane in 100 % O$_2$; Piramal Critical Care, Bethlehem, PA, USA) and received enrofloxacin (10 mg kg$^{-1}$ s.c.; Bayer Animal Health, Shawnee, KS, USA), buprenorphine (0.5 mg kg$^{-1}$ s.c.; Ceva Animal Health, Cambridge, ON, Canada) and warmed lactated Ringer's solution (5 mL s.c.; Baxter Corporation, Portland, OR, USA). The hind-paw withdrawal and the corneal reflexes were employed to test for an adequate anaesthesia level prior to commencing the surgery. Following T3 laminectomy, rats were trans-

ported and mounted on a plastic staging platform where the T2 and T4 spinous processes were stabilized with curved tip clamps. Contusion injury was induced using the custom impactor tip (3 mm; Infinite Horizons Impactor; Precision Systems and Instrumentation, Fairfax Station, VA, USA) dropped on the cord with predefined force with no dwell-time (force = 310 ± 12 kdyn; displacement = 1517 ± 177 microns; velocity = 122 ± 2 mm s$^{-1}$). Once injured, rats were recovered in an incubator for 30 min (37°C, 50% humidity) and received a subsequent 5 mL s.c. lactated Ringer's solution before they were returned to their home cages. The postoperative care for this experiment included administering s.c. lactated Ringer's solution (3× per day, 5 mL), buprenorphine (3× per day, 0.02 mg kg$^{-1}$) and enrofloxacin (1× per day, 10 mg kg$^{-1}$) up to for 4 days post-injury. Bladders were also manually expressed four times per day until spontaneous voiding resumed (4–6 days post-injury). During the recovery period, animals were provided with a supportive diet consisting of HydroGel (ClearH$_2$O, Westbrook, ME, USA), and melon, apple, spinach and cereal when necessary to maintain body mass during the postoperative period.

### General preparation for terminal experiments at 2 weeks post-SCI or in age-matched Naive animals

Animals were anaesthetized with isoflurane (5% isoflurane chamber induction and maintenance on 1.5%–2% isoflurane in 100% O$_2$; Piramal Critical Care) before conversion to urethane (up to a final dose of 2.1 g kg$^{-1}$ i.v.). Once a surgical plan of anaesthesia was reached using isoflurane [i.e. no hind-paw (for Naive animals) and front-paw (for SCI animals) withdrawal/corneal reflexes (for both Naive and SCI animals)], a solid-state pressure catheter (1.6F; Transonic, Ithaca, NY, USA) and a polyethylene catheter (PE-50 tubing; Intramedic™; BD, Franlin Lakes, NJ, USA) were then inserted in the femoral artery and vein, respectively, to continuously record arterial blood pressure (BP) and for i.v. fluid/drug infusion. Next, animals were intubated and mechanically ventilated (VentElite; Harvard Apparatus, Holliston, MA, USA). Breathing frequency and tidal volume for each animal were calculated using the previously described formulae (Pacher et al., 2008). Urethane was then delivered i.v. as a constant-rate infusion (6 mL h$^{-1}$) over ∼20–25 min (i.e. up to a final dose of 2.1 g kg$^{-1}$). Isoflurane was then slowly withdrawn during urethane infusion at the same time as BP, heart rate (HR), paw withdrawal and corneal reflexes were monitored to ensure an adequate anaesthesia level prior to commencing experiments. The adequacy of anaesthesia during surgical preparation was further tested by monitoring acute changes in BP (increase/decrease of 10 mmHg) in

response to noxious stimuli (tail or paw pinch), with maintenance doses of urethane infused as required (0.1 g kg$^{-1}$). The concentration of carbon dioxide was continuously monitored with a Gemini respiratory gas analyser (CWE Inc., Ardmore, PN, USA) and adjusted by changing the respiratory rate. Core temperature was maintained within a normal range (37.0 $\pm$ 0.5°C) using a servo-controlled heating pad (RightTemp® Temperature Monitor & Homeothermic Warming Control Module; Kent Scientific, Torrington, CT, USA).

### LV catheterization and data acquisition

The LV catheterization was performed via a closed-chest, right common carotid artery approach using a pressure–volume admittance catheter (1.9F; Transonic). We considered this approach over open-chest since this method maintains intrathoracic pressure and its ensuing impacts on cardiorespiratory function. Briefly, the right common carotid artery was isolated and temporarily occluded using rostral and caudal silk sutures. A small hole was pierced on the ventral aspect of the artery with a customized tool (bent 25 gauge needle) and an admittance catheter advanced into the artery. The catheter was then advanced, under pressure guidance into the LV. Following a 30 min haemodynamic stabilization period, baseline LV indices were assessed. A solid-state pressure catheter (2F; Transonic) was also inserted in a femoral artery to allow for continuous monitoring of arterial BP. Transducers were connected in series to an analogue-to-digital converter (1 kHz s$^{-1}$ sampling rate; PowerLab; ADInstruments, Sydney, NSW, Australia), which allowed for real-time monitoring of cardiovascular function via commercially available software (LabChart, version 8.1; AD Instruments). For each rat, we allowed for a 10 min period of baseline recordings, the last 60 s of which was used to extract and calculate beat-by-beat LV and arterial indices.

### AIH protocol

For hypoxia delivery, we used commercially available small rodent ventilation (VentElite, Harvard Apparatus) and a computer-controlled gas mixer (CWE Inc.) to expose animals to 10 bouts of %10 O$_2$ (balanced with nitrogen) lasting for 1 min interspersed with 2 min of breathing %100 O$_2$. This protocol is a modified iteration of the hypoxia protocol that has previously shown to effectively elicit sympathetic LTF in intact rodents (Dick et al., 2007) and is the identical protocol that we used in our recent study demonstrating AIH induces robust sLTF in rodents with SCI (Ahmadian et al., 2025).

### Animal death

After completing the terminal procedures and data collection (i.e. animals did not ever gain consciousness), animals remained under urethane anaesthesia (i.e. up to a final dose of 2.1 g kg$^{-1}$ I.P.) and were then killed via chloral hydrate (1 g kg$^{-1}$ I.P.), followed by thoracotomy and exsanguination via cardiac puncture. Prior to injecting chloral hydrate, animals were carefully examined for the depth of anaesthesia using a combination of checking for paw withdrawal and corneal reflexes, as well as ensuring the acute changes in BP in response to noxious stimuli did not exceed 10 mmHg. Death was confirmed by cessation of breathing for at least 3 min upon removal of animals from the ventilator and by viewing the real-time arterial pressure signals from the catheter that remained *in situ* (i.e. confirming the permanent cessation of the circulation).

### Statistical analysis

The Shapiro–Wilk test was used to assess distribution characteristics. For experiment 1 (cardiovascular responses to autonomic blockades in SCI *vs.* Naive animals) and experiment 2 (cardiovascular functional benefits of AIH in SCI animals), statistical analysis was performed using a two-way (group × time) mixed analysis of variance, with Bonferroni correction for multiple comparison testing. Note that, although for experiment 2, we present our results with respect to the blockade condition for brevity, we used a single statistical model to interrogate changes in cardiac (vascular) function across all blockade conditions. For experiment 1, we based our sample size calculation on our prior work examining the effects of hexamethonium bromide on cardiac function in rats following acute high-thoracic complete SCI (Fossey et al., 2022). A total sample size of at least four animals was required to detect a significant difference/reduction ($P < 0.05$, 95% power) in the pressure-generating capacity of the heart (i.e. d$P$/d$t_{max}$) post-blockade *vs.* baseline. For experiment 2, we based our sample size calculation on prior work examining AIH (10× F$_i$O$_2$ = 0.07–0.08) and systemic blood pressure in Naive rats (Lemes et al., 2016). We found that a total sample size of at least four animals was required to detect a significant within-group (pre-AIH *vs.* post-AIH) difference ($P < 0.05$, 95% power) for a 10 mmHg increase of mean arterial pressure (MAP). However, we used 8–12 rats to account for the design of our study (i.e. the two-way design of the present study *vs.* the within animal design of the prior study), the potential mortality associated with SCI surgery, post-SCI care and the invasive cardiovascular surgeries that were employed. The exact numbers of animals used in the final analyses for each experiment are mentioned where appropriate. Data analyses were performed using Prism, version 9.1.2

(GraphPad Software Inc., San Diego, CA, USA) and figures were created using BioRender.com. For brevity, we have focused our figures on key indices of heart function (LV $dP/dt_{max}$) and diastolic blood pressure (DBP), as well as HR. All measured indices are reported in the associated tables. Data are presented as the mean ± SD. The alpha value was set at $P < 0.05$.

## Results

### Effects of sympathetic blockade on cardiovascular function following SCI

As expected, we found that SCI was associated with significant reductions in heart function ($dP/dt_{max}$, $P < 0.001$) and blood pressure (i.e. DBP, $P = 0.002$) compared to Naive animals. To block sympathetic transmission to the heart, we administered the cardio-selective beta-1 adrenergic receptor antagonist esmolol (Fig. 1*A*). Esmolol caused a significant reduction in $dP/dt_{max}$, HR and DBP in animals with (all $P < 0.001$) and without SCI (all $P < 0.001$) relative to baseline (Fig. 1*B* and *Ca–Ea* and Table 1). Although the magnitude of the drop for HR did not differ between groups ($P = 0.13$), there was a significantly greater drop for the pressure-generating capacity of the heart (i.e. $dP/dt_{max}$, $P < 0.001$) and systemic arterial pressure (i.e. DBP, $P = 0.05$) in Naive rats in response to esmolol (Fig. 1*Cb–Eb* and Table 2). Indeed, esmolol infusion in Naive animals resulted in a reduction in $dP/dt_{max}$ to values that were no longer different to the SCI values observed at baseline.

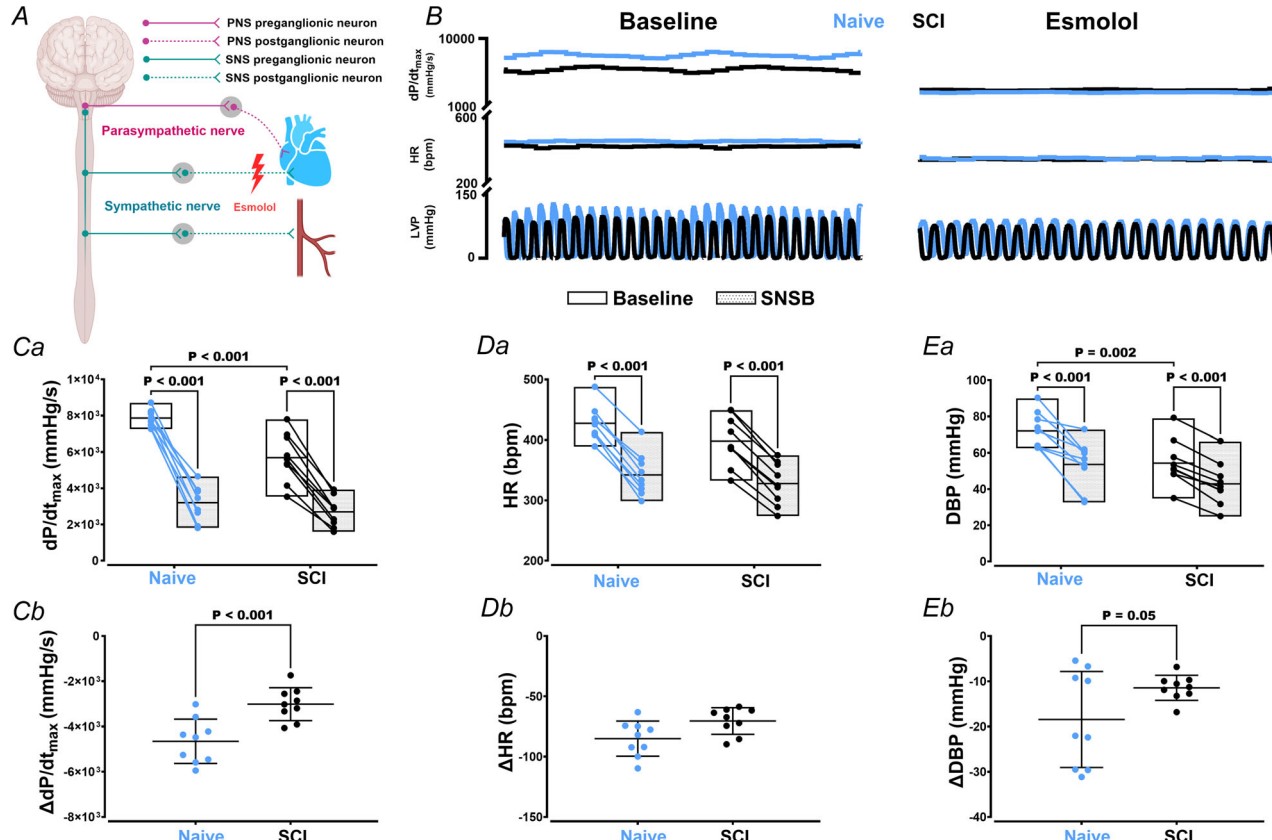

**Figure 1. Cardiovascular response to sympathetic blockade with esmolol in animals with (*n* = 9) and without (*n* = 9) spinal cord injury (SCI)**
*A*, neuroanatomical overview. *B*, raw traces of left ventricular pressure (LVP), heart rate (HR) and the maximal rate of rise of left ventricular pressure ($dP/dt_{max}$) during baseline and sympathetic blockade with esmolol. Group data for $dP/dt_{max}$ (*Ca*), HR (*Da*) and DBP (*Ea*) during baseline and sympathetic blockade with esmolol. *Cb–Eb*, the magnitude of changes from baseline to blockade condition. Note that, although there was a reduction in the pressure-generating of the heart and systemic arterial pressure in both Naive and SCI animals, the magnitude of the drop in $dP/dt_{max}$ was greater in SCI animals. Statistical analysis was performed using a two-way (group × time) mixed analysis of variance, with Bonferroni correction for multiple-comparison testing. PNS, parasympathetic nervous system; SNS, sympathetic nervous system. SNSB, SNS blockade. [Colour figure can be viewed at wileyonlinelibrary.com]

**Table 1. Cardiovascular function in response to autonomic blockade in animals with ($n = 9$) and without ($n = 9$) high-thoracic spinal cord injury (SCI).**

| Variables | Baseline | | SNSB | | PNSB | | DB | | CB | |
|---|---|---|---|---|---|---|---|---|---|---|
| | Naive | SCI | Naive | SCI | Naive | SCI | Naive | SCI | Naive | SCI |
| **LV systolic function measures** | | | | | | | | | | |
| $P_{max}$ (mmHg) | 118 ± 10 | 93 ± 14## | 87 ± 17** | 74 ± 16*** | 111 ± 9 | 94 ± 19 | 88 ± 14*** | 80 ± 19** | 53 ± 7*** | 70 ± 16*** |
| $dP/dt_{max}$ (mmHg s$^{-1}$) | 7866 ± 488 | 5703 ± 1349## | 3211 ± 976*** | 2691 ± 812*** | 7409 ± 834 | 5706 ± 1566 | 3210 ± 829*** | 2979 ± 1000*** | 1725 ± 266*** | 2369 ± 716*** |
| P@$dP/dt_{max}$ (mmHg) | 57 ± 4 | 43 ± 7## | 38 ± 12** | 31 ± 9*** | 53 ± 5** | 43 ± 10 | 38 ± 8*** | 34 ± 10*** | 21 ± 5*** | 28 ± 8*** |
| CPP (mmHg) | 68 ± 10 | 53 ± 12 | 47 ± 15** | 40 ± 12** | 63 ± 13 | 53 ± 14 | 49 ± 15** | 43 ± 14*** | 23 ± 8*** | 34 ± 9** |
| RPP (mmHg-bpm) | 49,271 ± 5383 | 38,343 ± 6630## | 30,110 ± 6505*** | 25,491 ± 5326*** | 47,592 ± 8436 | 38,583 ± 7014 | 31,069 ± 6780*** | 27,552 ± 5960*** | 15,312 ± 2878*** | 20,265 ± 4637*** |
| **LV diastolic function measures** | | | | | | | | | | |
| $dP/dt_{min}$ (mmHg s$^{-1}$) | −5046 ± 596 | −4107 ± 810 | −3024 ± 1116** | −2549 ± 922*** | −4529 ± 721* | −4160 ± 1212 | −3052 ± 962*** | −2858 ± 1043*** | −1340 ± 404*** | −2100 ± 751*** |
| EDP (mmHg) | 4 ± 2.1 | 1.38 ± 2.1 | 6.6 ± 3.2 | 2.8 ± 2.2 | 3.4 ± 2.5 | 1.4 ± 2.6 | 6.3 ± 3.7 | 3 ± 2.3 | 4.1 ± 2.7 | 2.9 ± 2 |
| **Systemic arterial pressure** | | | | | | | | | | |
| SBP (mmHg) | 116 ± 14 | 91 ± 14# | 89 ± 21** | 74 ± 17*** | 109 ± 16* | 91 ± 19 | 90 ± 19*** | 79 ± 20** | 48 ± 6*** | 65 ± 14*** |
| DBP (mmHg) | 72 ± 10 | 55 ± 13# | 54 ± 14** | 43 ± 12** | 66 ± 12* | 54 ± 15 | 55 ± 13** | 56 ± 14** | 27 ± 6*** | 38 ± 9** |
| MAP (mmHg) | 89 ± 10 | 70 ± 14# | 69 ± 16** | 57 ± 15** | 83 ± 13** | 71 ± 18 | 70 ± 15*** | 61 ± 18** | 35 ± 6*** | 49 ± 12***# |
| PP (mmHg) | 43 ± 8 | 36 ± 9 | 35 ± 10* | 31 ± 8* | 43 ± 7 | 37 ± 8 | 35 ± 8** | 33 ± 8 | 21 ± 3*** | 28 ± 7*** |
| HR (bpm) | 427 ± 29 | 399 ± 41 | 342 ± 35*** | 328 ± 35*** | 434 ± 23 | 395 ± 36 | 347 ± 28*** | 323 ± 35*** | 317 ± 36*** | 298 ± 37*** |

*Note*: Data are presented as the mean ± SD. Statistical analysis was performed using a two-way (group × time) mixed analysis of variance, with Bonferroni correction for multiple-comparison testing. Estimated CPP is calculated as the difference between DBP and ventricular EDP (Nguyen et al., 2018). RPP is calculated as the product of HR and SBP (Ansari et al., 2012). Abbreviations: CB, complete blockade; CPP, coronary perfusion pressure; DB, double blockade; DBP, diastolic blood pressure; $dP/dt_{max}$, the maximal rate of rise of the LV pressure; $dP/dt_{min}$, the maximal rate of decrease of the LV pressure; EDP, end-diastolic pressure; HR, heart rate.LV, left ventricle; MAP, mean arterial pressure; P@$dP/dt_{max}$, pressure at $dP/dt_{max}$; $P_{max}$, maximal pressure of the LV pressure; PNSB, parasympathetic nervous system blockade; PP, pulse pressure; RPP, rate pressure product; SBP, systolic blood pressure; SNSB, sympathetic nervous system blockade. *$P < 0.05$, **$P < 0.05$, ***$P < 0.001$ *vs.* baseline in the same experimental group. #$P < 0.05$, ##$P < 0.05$, ###$P < 0.001$ *vs.* Naive in same phase.

Table 2. The magnitude of changes from baseline in cardiovascular function during each blockade trial in animals with ($n = 9$) and without ($n = 9$) high-thoracic spinal cord injury (SCI)

| Variables | ΔSNSB | | | ΔPNSB | | | ΔDB | | | ΔCB | | | ΔCB-DB | | |
|---|---|---|---|---|---|---|---|---|---|---|---|---|---|---|---|
| | Naïve | SCI | P | Naïve | SCI | P | Naïve | SCI | P | Naïve | SCI | P | Naïve | SCI | P |
| **LV systolic function measures** | | | | | | | | | | | | | | | |
| $P_{max}$ (mmHg) | −31 ± 19 | −19 ± 5 | 0.478 | −7 ± 7 | 1 ± 8 | 0.114 | −30 ± 15 | −13 ± 7 | 0.050 | −65 ± 8 | −23 ± 5 | **<0.001** | −35 ± 8 | −10 ± 7 | **<0.001** |
| $dP/dt_{max}$ (mmHg s⁻¹) | −4655 ± 981 | −3013 ± 727 | **0.006** | −457 ± 740 | 2 ± 652 | 0.908 | −4657 ± 866 | −2724 ± 649 | **<0.001** | −6141 ± 495 | −3334 ± 855 | **<0.001** | −1485 ± 597 | −610 ± 436 | **0.015** |
| $P@dP/dt_{max}$ (mmHg) | −20 ± 10 | −12 ± 2 | 0.206 | −4 ± 2 | −0.1 ± 4 | 0.167 | −19 ± 5 | −9 ± 4 | **0.001** | −36 ± 2 | −15 ± 3 | **<0.001** | −17 ± 4 | −6 ± 4 | **<0.001** |
| CPP (mmHg) | −21 ± 11 | −13 ± 3 | 0.329 | −6 ± 6 | −0.5 ± 6 | 0.398 | −20 ± 10 | −11 ± 4 | 0.128 | −45 ± 5 | −19 ± 10 | **<0.001** | −26 ± 8 | −9 ± 9 | **0.003** |
| RPP (mmHg·bpm) | −19,161 ± 4852 | −11,929 ± 1144 | **0.009** | −1679 ± 4037 | 23 ± 4295 | >0.999 | −18,202 ± 3848 | −10,445 ± 3309 | **0.001** | −33,959 ± 3173 | −16,605 ± 3844 | **<0.001** | −15,757 ± 5203 | −6161 ± 4551 | **0.003** |
| **LV diastolic function measures** | | | | | | | | | | | | | | | |
| $dP/dt_{min}$ (mmHg s⁻¹) | 2022 ± 1114 | 1558 ± 446 | >0.999 | 517 ± 436 | −53 ± 586 | 0.168 | 1994 ± 925 | 1250 ± 498 | 0.272 | 3706 ± 502 | 2007 ± 392 | **<0.001** | 1712 ± 594 | 757 ± 461 | **0.008** |
| EDP (mmHg) | 2.6 ± 2.1 | 1.4 ± 0.7 | 0.671 | −0.6 ± 1.2 | 0.04 ± 1.3 | >0.999 | 2.3 ± 2.7 | 1.7 ± 1 | >0.999 | 0.1 ± 1.9 | 1.5 ± 0.7 | 0.288 | −2.2 ± 1.3 | −0.1 ± 1.1 | **0.016** |
| **Systemic arterial pressure** | | | | | | | | | | | | | | | |
| SBP (mmHg) | −27 ± 16 | −17 ± 5 | 0.531 | −6 ± 5 | 1 ± 7 | 0.177 | −26 ± 9 | −12 ± 8 | **0.015** | −67 ± 9 | −26 ± 9 | **<0.001** | −41 ± 14 | −14 ± 13 | **0.002** |
| DBP (mmHg) | −18 ± 10 | −11 ± 3 | 0.438 | −6 ± 5 | −0.4 ± 7 | 0.293 | −18 ± 8 | −9 ± 5 | 0.073 | −45 ± 5 | −18 ± 10 | **<0.001** | −28 ± 8 | −9 ± 10 | **0.001** |
| MAP (mmHg) | −20 ± 13 | −14 ± 3 | 0.856 | −6 ± 4 | 0.2 ± 7 | 0.208 | −19 ± 9 | −10 ± 6 | 0.086 | −54 ± 5 | −21 ± 10 | **<0.001** | −35 ± 10 | −12 ± 11 | **0.001** |
| PP (mmHg) | −8 ± 7 | −5 ± 4 | >0.999 | −0.1 ± 4 | 1 ± 4 | >0.999 | −9 ± 5 | −3 ± 6 | 0.166 | −22 ± 9 | −8 ± 3 | **0.005** | −14 ± 8 | −5 ± 3 | 0.071 |
| HR (bpm) | −85 ± 15 | −70 ± 11 | 0.148 | 6 ± 23 | −4 ± 23 | >0.999 | −80 ± 22 | −76 ± 24 | >0.999 | −111 ± 29 | −101 ± 20 | >0.999 | −30 ± 14 | −25 ± 12 | >0.999 |

Data are presented as the mean ± SD. Statistical analysis was performed using a two-way (group × time) mixed analysis of variance, with Bonferroni correction for multiple-comparison testing. *P* values in bold are significant observations. Estimated CPP is calculated as the difference between DBP and ventricular EDP (Nguyen et al., 2018). RPP is calculated as the product of HR and SBP (Ansari et al., 2012). Abbreviations: CB, complete blockade; CPP, coronary perfusion pressure; DB, double blockade; DBP, diastolic blood pressure; $dP/dt_{max}$, the maximal rate of rise of the LV pressure; $dP/dt_{min}$, the maximal rate of decrease of the LV pressure; EDP, end-diastolic pressure; HR, heart rate; LV, left ventricle; MAP, mean arterial pressure; $P@dP/dt_{max}$, pressure at $dP/dt_{max}$; $P_{max}$, maximal pressure of the LV pressure; PNSB, parasympathetic nervous system blockade; PP, pulse pressure; RPP, rate pressure product; SBP, systolic blood pressure; SNSB, sympathetic nervous system blockade.

## Effects of parasympathetic blockade on cardiovascular function following SCI

To explore the sole impact of parasympathetic communication on the heart, we waited for esmolol to wash out and then administered the muscarinic receptor antagonist atropine (Fig. 2*A*). We observed no significant changes in d$P$/d$t_{max}$, HR and DBP in both animals with (all $P > 0.05$) and without SCI (all $P > 0.05$) relative to baseline (Fig. 2*B* and *Ca–Ea* and Table 1). Similarly, the magnitude of drop for these metrics also did not differ significantly (all $P > 0.05$) between groups (Fig. 2*Cb–Eb* and Table 2), with reduced cardiovascular function still persisting in animals with SCI *vs.* Naive animals.

## Effects of combined sympathetic and parasympathetic (double) blockade on cardiovascular function following SCI

We next re-infused esmolol, in addition to continuing to infuse atropine, to block both sympathetic and parasympathetic transmission to the heart (i.e. complete cardiac autonomic blockade) (Fig. 3*A*). We observed a significant reduction in d$P$/d$t_{max}$, HR and DBP in both animals with (all $P < 0.001$) and without SCI (all $P < 0.05$) relative to baseline (Fig. 3*B* and *Ca–Ea* and Table 1). Although the magnitude of the drop for HR ($P = 0.64$) did not differ between groups, it was significantly greater for the pressure-generating capacity of the heart (i.e. d$P$/d$t_{max}$, $P < 0.001$) and systemic arterial pressure (i.e. DBP, $P = 0.01$) in Naive rats (Fig. 3*Cb–Eb* and Table 2).

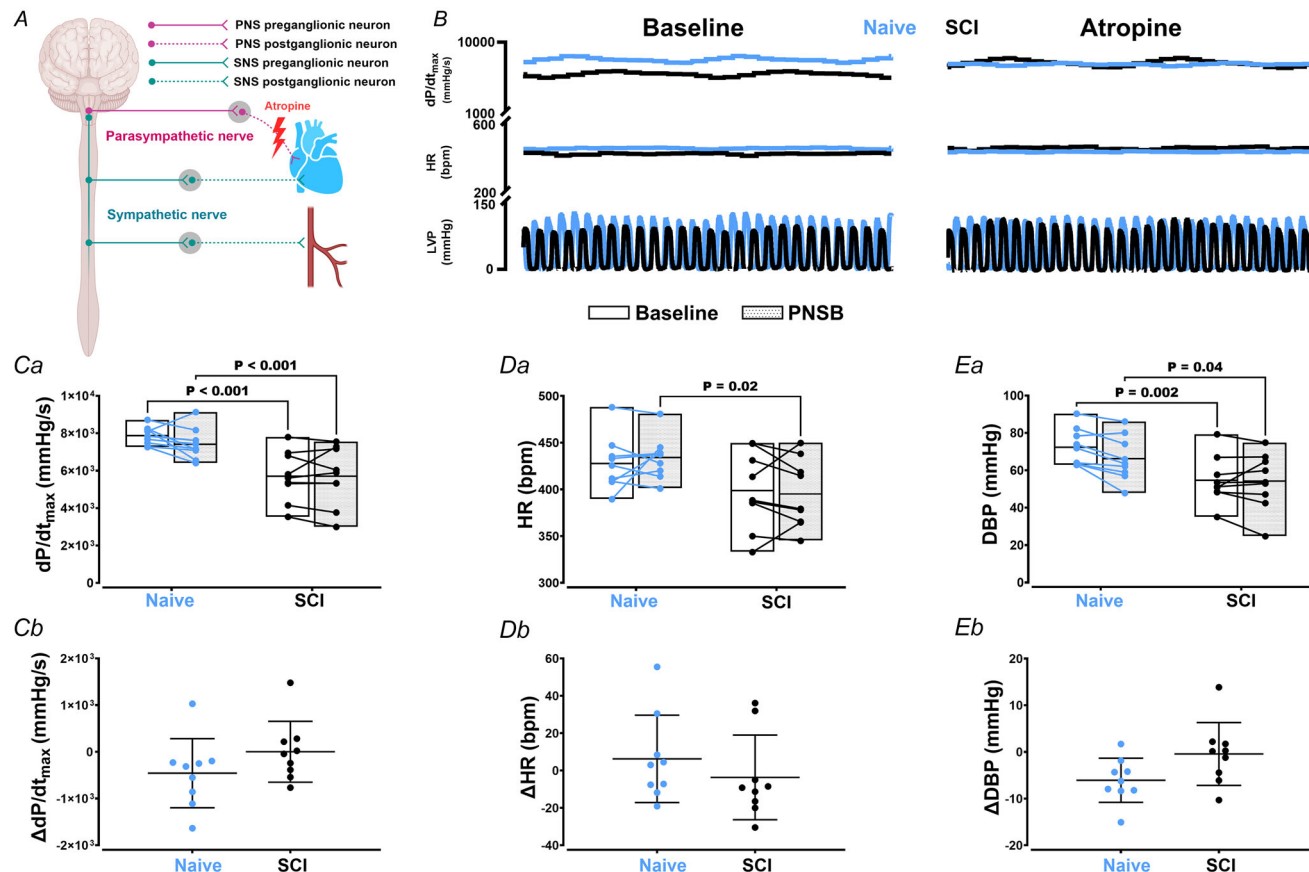

**Figure 2. Cardiovascular response to parasympathetic blockade with atropine in animals with (*n* = 9) and without (*n* = 9) spinal cord injury (SCI)**

*A*, neuroanatomical overview. *B*, raw traces of left ventricular pressure (LVP), heart rate (HR) and the maximal rate of rise of left ventricular pressure (d$P$/d$t_{max}$) during baseline and parasympathetic blockade with atropine. Group data for d$P$/d$t_{max}$ (*Ca*), HR (*Da*) and DBP (*Ea*) during baseline and parasympathetic blockade with atropine. *Cb–Eb*, the magnitude of changes from baseline to blockade condition. Note that the reduced cardiovascular function in SCI *vs.* Naive animals is still detectable under this blockade condition. Statistical analysis was performed using a two-way (group × time) mixed analysis of variance, with Bonferroni correction for multiple-comparison testing. Note that, although there was a reduction in the pressure-generating of the heart and systemic arterial pressure in both Naive and SCI animals, the magnitude of the drop in d$P$/d$t_{max}$ was greater in SCI animals. PNS, parasympathetic nervous system; SNS, sympathetic nervous system. PNSB, PNS blockade. [Colour figure can be viewed at wileyonlinelibrary.com]

Combined esmolol and atropine infusion in Naive animals resulted in a reduction in d$P$/d$t_{max}$ to values that were no longer different to the SCI values observed at baseline.

## Vascular influence on cardiac function following SCI

We next added hexamethonium bromide to the double blockade condition to examine additional vascular influences on cardiac dysfunction post-SCI (Fig. 4$A$). We observed an additional, concurrent significant reduction in the pressure-generating capacity of the heart (i.e. d$P$/d$t_{max}$, $P < 0.001$) and systemic arterial pressure (i.e. DBP, $P < 0.001$) in Naive animals (Fig. 4$B$, $Ca$ and $Da$ and Table 1). However, unlike Naive rats, a significant reduction in DBP was not accompanied by a significant reduction in d$P$/d$t_{max}$ in SCI rats (Fig. 4$B$, $Ca$ and $Da$). The magnitude of drops for these metrics (d$P$/d$t_{max}$,

$P = 0.01$; DBP, $P < 0.001$) was also significantly less for SCI *vs.* Naive rats (Fig. 4$Cb$ and $Db$ and Table 2). We also found a positive and statistically significant relationship between the drops in the DBP and drops in d$P$/d$t_{max}$ in both animal groups (Fig. 4$E$). Although reductions in systemic arterial pressure (i.e. DBP) were associated with a significant reduction in heart function (i.e. d$P$/d$t_{max}$) in both groups, this link appears to be notably less pronounced in SCI rats, evidenced by a lower regression slope for SCI *vs.* Naive (42 *vs.* 67) animals (Fig. 4$E$).

## Cardiovascular function in response to AIH following SCI

To explore whether AIH is able to neuromodulate the heart post-SCI, two groups of chronic SCI rats (AIH and TC) were instrumented with cardiac and arterial catheters

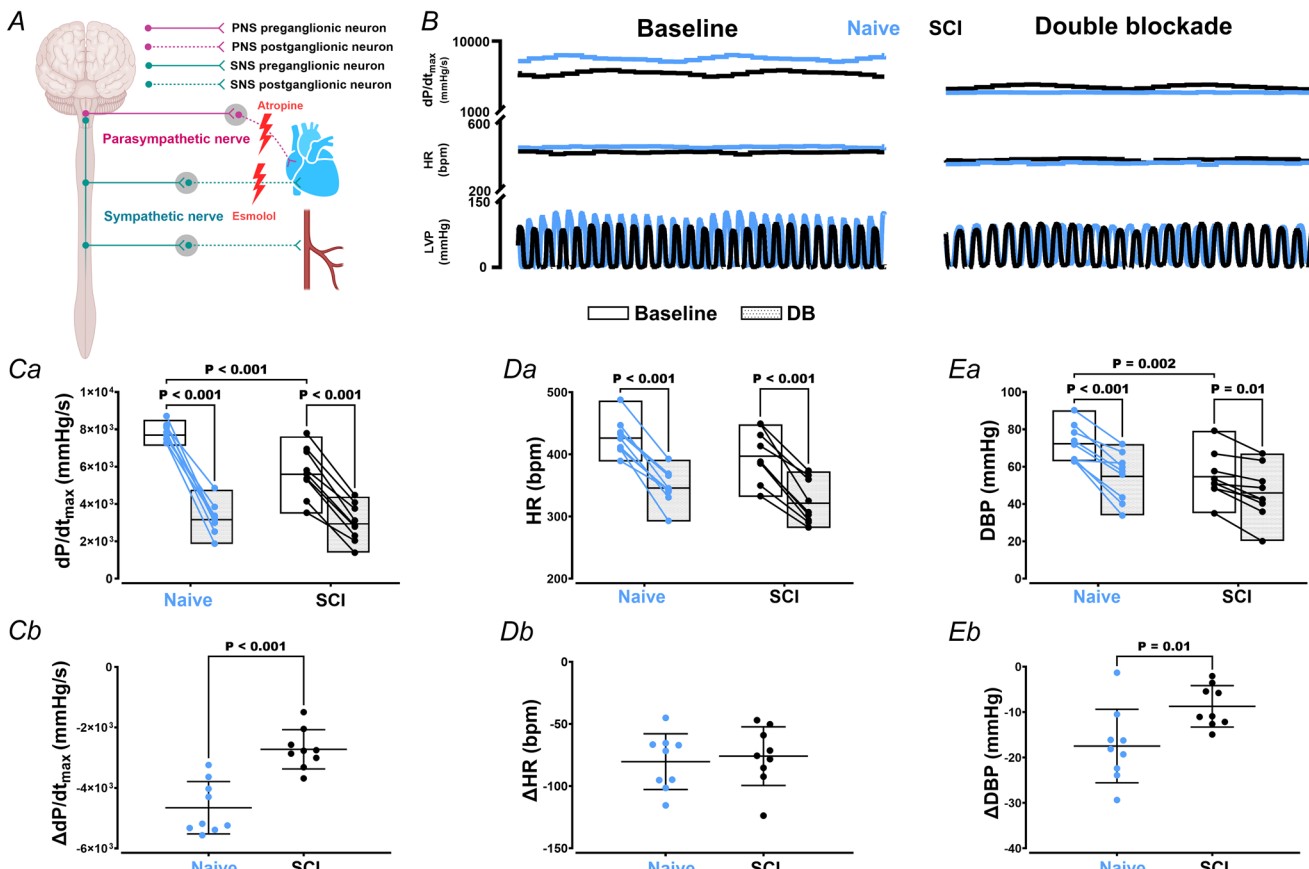

**Figure 3. Cardiovascular response to double cardiac autonomic blockade (esmolol and atropine) in animals with (*n* = 9) and without (*n* = 9) spinal cord injury (SCI)**
*A*, neuroanatomical overview. *B*, raw traces of left ventricular pressure (LVP), heart rate (HR), and the maximal rate of rise of left ventricular pressure (d$P$/d$t_{max}$) during baseline and double blockade (DB) with esmolol and atropine. Group data for d$P$/d$t_{max}$ (*Ca*), HR (*Da*) and DBP (*Ea*) during baseline and DB. *Cb–Eb*, the magnitude of changes from baseline to blockade condition. Note the greater drop in the pressure-generating of the heart and systemic arterial pressure in Naive *vs.* SCI animals, as well as equalization of cardiovascular function between these animals under DB condition. Statistical analysis was performed using a two-way (group × time) mixed analysis of variance, with Bonferroni correction for multiple-comparison testing. PNS, parasympathetic nervous system; SNS, sympathetic nervous system. [Colour figure can be viewed at wileyonlinelibrary.com]

and assessed for cardiovascular function at baseline and different time points [immediately when the final hypoxic bout was over (1 min) and 90 min] post-exposure (Fig. 5*A* and *B*). Relative to baseline, metrics associated with cardiac pressure-generating capacity and representative of systolic function (e.g. $P_{max}$, $dP/dt_{max}$, rate pressure product), together with diastolic function (i.e. $dP/dt_{min}$) and systemic arterial pressure [e.g. DBP, systolic blood pressure (SBP) and MAP], were increased immediately after and up to 90 min post-AIH *vs*. TC rats (Fig. 5*Ca*–*Ea* and Table 3). Such AIH-induced increases in cardiac and systemic pressures were further confirmed by observing the greater magnitude of change from baseline in AIH rats when compared to TC rats receiving no AIH

(Fig. 5*Cb*–*Eb* and Table 4). Augmented cardiovascular function occurred in concert with increased estimated coronary perfusion pressure (i.e. DBP – ventricular end-diastolic pressure) (Fig. 5*Fa* and *Fb* and Tables 3 and 4). Note that baseline between-group (i.e. TC *vs*. AIH) differences were noted for a few metrics (i.e. $P_{max}$, SBP, pulse pressure and HR).

## Discussion

The comprehensive pharmacological investigation(s) performed during these studies identified the loss/weakened sympathetic control as the principal precipitant of cardiac (vascular) dysfunction post-SCI.

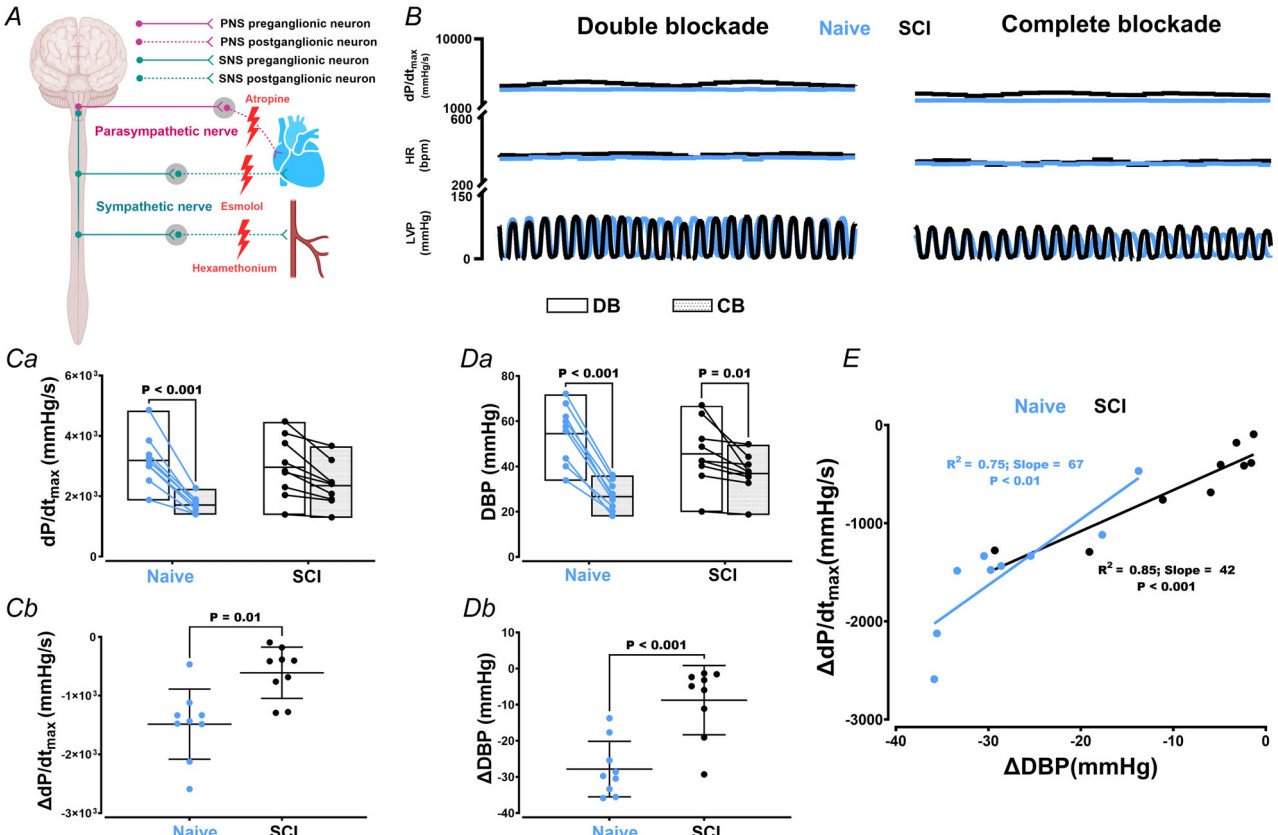

**Figure 4. Vascular influences on cardiac function in animals with (*n* = 9) and without (*n* = 9) spinal cord injury (SCI)**

*A*, neuroanatomical overview. *B*, raw traces of left ventricular pressure (LVP), heart rate (HR) and the maximal rate of rise of left ventricular pressure ($dP/dt_{max}$) during double (DB; esmolol and atropine) and complete (CB; esmolol, atropine, and hexamethonium) blockade. Group data for $dP/dt_{max}$ (*Ca*) and DBP (*Da*) during DB and CB. *Cb*–*Db*, the magnitude of changes from DB to CB condition. *E*, association between the drops in systemic arterial pressure (i.e. DBP) and heart function (i.e. $dP/dt_{max}$) in Naive and SCI rats. Note that, unlike Naive rats, the greater drops in systemic arterial pressure did not occur in concert with drops in the pressure-generating capacity of the heart in SCI rats when transitioning from DB to CB, with SCI rats also showing a weaker association between the drops in systemic arterial pressure and heart function. Statistical analysis was performed using a two-way (group × time) mixed analysis of variance, with Bonferroni correction for multiple-comparison testing. The association between the drops in systemic arterial pressure and heart function was examined using a linear regression analysis. For the regression analysis, the delta for metrics of interest was calculated as the difference between CB and DB conditions. PNS, parasympathetic nervous system; SNS, sympathetic nervous system. [Colour figure can be viewed at wileyonlinelibrary.com]

**Table 3. Cardiovascular function prior to and following AIH delivery across different experimental groups**

| Variables | TC ($n = 8$) | | | | AIH ($n = 11$) | | | |
|---|---|---|---|---|---|---|---|---|
| | Baseline | Post | 90min | P | Baseline | Post | 90 min | P |
| **LV systolic function measures** | | | | | | | | |
| $P_{max}$ (mmHg) | 112 ± 4 | 114 ± 6 | 114 ± 5 | NS | 95 ± 12### | 104 ± 13 | 106 ± 9 | 90 min***, Post** > Baseline |
| $dP/dt_{max}$ (mmHg s$^{-1}$) | 6724 ± 746 | 6822 ± 686 | 6822 ± 635 | NS | 6217 ± 884 | 7114 ± 1305 | 7263 ± 1254 | 90 min**, Post** > Baseline |
| $P@dP/dt_{max}$ (mmHg) | 57 ± 3 | 57 ± 4 | 57 ± 3 | NS | 53 ± 7 | 58 ± 7 | 58 ± 4 | 90 min**, Post** > Baseline |
| CPP (mmHg) | 60 ± 7 | 60 ± 6 | 60 ± 7 | NS | 56 ± 8 | 63 ± 10 | 62 ± 6 | 90 min**, Post** > Baseline |
| RPP (mmHg·bpm) | 42,680 ± 3791 | 42,290 ± 4024 | 42,156 ± 3895 | NS | 42,918 ± 7447 | 48,664 ± 9186 | 47,960 ± 6763 | 90 min**, Post** > Baseline |
| **LV diastolic function measures** | | | | | | | | |
| $dP/dt_{min}$ (mmHg s$^{-1}$) | −5278 ± 403 | −5218 ± 455 | −5246 ± 449 | NS | −4324 ± 615 | −5061 ± 1003 | −5073 ± 722 | 90 min***, Post*** > Baseline |
| EDP (mmHg) | 7.5 ± 1.5 | 7.5 ± 1.7 | 7.4 ± 2.8 | NS | 5.2 ± 3.7 | 4.9 ± 2.4 | 4.8 ± 2.8 | ns |
| **Systemic arterial pressure** | | | | | | | | |
| SBP (mmHg) | 112 ± 11 | 114 ± 11 | 114 ± 11 | NS | 95 ± 12### | 104 ± 13 | 106 ± 9 | 90 min***, Post** > Baseline |
| DBP (mmHg) | 68 ± 7 | 67 ± 5 | 67 ± 6 | NS | 61 ± 10 | 67 ± 11 | 67 ± 7 | 90 min*, Post* > Baseline |
| MAP (mmHg) | 83 ± 5 | 83 ± 4 | 83 ± 5 | NS | 72 ± 10 | 80 ± 11 | 80 ± 7 | 90 min**, Post** > Baseline |
| PP (mmHg) | 45 ± 8 | 47 ± 8 | 47 ± 7 | NS | 34 ± 5## | 37 ± 6## | 39 ± 6 # | 90 min**, Post** > Baseline |
| HR (bpm) | 380 ± 32 | 370 ± 33 | 369 ± 36 | NS | 450 ± 49## | 465 ± 52### | 453 ± 61### | 90 min*** > Baseline, NS |

*Note*: Data are presented as the mean ± SD. Statistical analysis was performed using a two-way (group × time) mixed analysis of variance, with Bonferroni correction for multiple-comparison testing. Estimated CPP is calculated as the difference between DBP and ventricular EDP (Nguyen et al., 2018). RPP is calculated as the product of HR and SBP (Ansari et al., 2012).

Abbreviations:

AIH, hypoxia-exposed group; CPP, coronary perfusion pressure; DBP, diastolic blood pressure; $dP/dt_{max}$, the maximal rate of rise of the LV pressure; $dP/dt_{min}$, the maximal rate of decrease of the LV pressure; EDP, end-diastolic pressure; HR, heart rate; LV, left ventricle; MAP, mean arterial pressure; NS, not significant; $P@dP/dt_{max}$, pressure at $dP/dt_{max}$; $P_{max}$, maximal pressure of the LV pressure; PP, pulse pressure; RPP, rate pressure product; SBP, systolic blood pressure; TC, time control group.
*$P < 0.05$, **$P < 0.05$, ***$P < 0.001$ *vs.* baseline in the same group. #$P < 0.05$, ##$P < 0.05$, ###$P < 0.001$ *vs.* TC in the same phase.

We found that parasympathetic control remains intact and does not contribute to cardiac dysfunction post-SCI. In the second part of the present study, we provide the first evidence that AIH can neuromodulate the heart post-SCI.

We found that sympathetic and double blockade resulted in significant drops in the pressure-generating capacity of the heart (i.e. $dP/dt_{max}$), chronotropic function (i.e. HR), and systemic arterial pressure (i.e. DBP) *vs.* baseline in both Naive animals and animals with SCI. We found a significantly greater magnitude of reduction for $dP/dt_{max}$ and DBP with both sympathetic and double blockade for Naive rats, which effectively equalized the cardiovascular function between animals with and without SCI. The smaller reduction in $dP/dt_{max}$ and DBP for SCI animals, as well as the ensuing elimination of baseline group differences, experimentally confirm the key regulatory role of sympathetic control in cardiovascular dysfunction in our rat model of high-thoracic

SCI. Such experimental observations extend findings from previous studies, which suggested that the partial/full (i.e. high-thoracic/cervical injuries) loss of bulbospinal sympathetic control precludes normal cardiovascular function/regulation post-injury (Fossey et al., 2022; Lujan & DiCarlo, 2020; Lujan et al., 2018).

Previous experimental evidence suggests that the LV, although primarily under sympathetic control, is also under a tonic inhibitory muscarinic influence (Machhada et al., 2016). In the setting of experimental SCI, mid-thoracic SCI has been previously reported to cause structural remodelling of the heart (e.g. LV chamber size, wall thickness, collagen content) and vagal preganglionic neurons of the nucleus ambiguous (e.g. dendritic arborization, morphology) in rats (Lujan et al., 2014). Although this experimental evidence clearly associates structural remodelling of vagal preganglionic neurons with heart function post-SCI, to our knowledge, no studies (either in animal models or

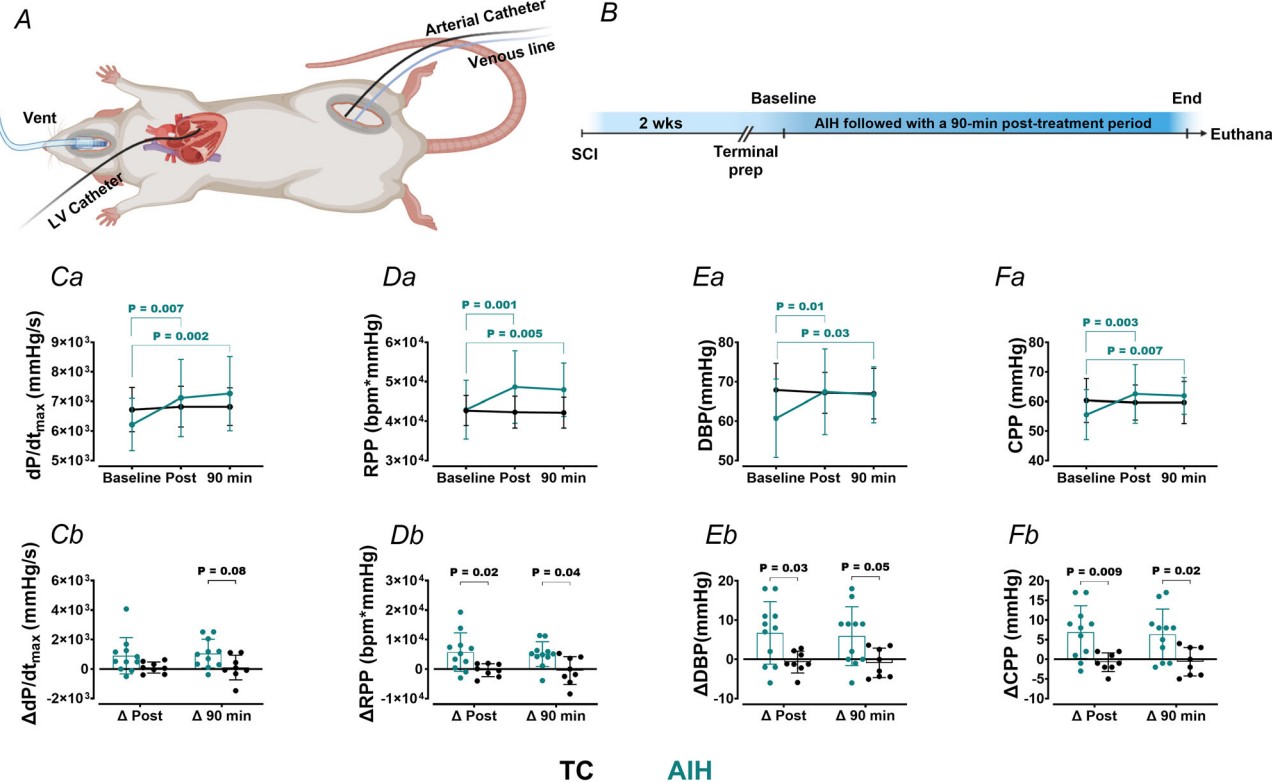

**Figure 5. Cardiovascular responses to acute intermittent hypoxia (AIH) and time-control following spinal cord injury (SCI)**

*A*, terminal preparation and *B*, experimental design used. Note that the maximal rate of rise of LV pressure (*Ca*, $dP/dt_{max}$), rate pressure product (*Da*, RPP), diastolic blood pressure (*Ea*, DBP), and coronary perfusion pressure (*Fa*, CPP) were augmented immediately after and up to 90 min post-AIH (10 × 1 min episodes; $F_{IO_2} = 0.1$, interspersed with 2 min of $F_{IO_2} = 1.0$) in AIH-exposed animals (AIH, $n = 11$) *vs.* time control animals (TC, $n = 8$) without hypoxia. *Cb–Fb*, showing changes in metrics between each time point and baseline. Statistical analysis was performed using a two-way (group × time) mixed analysis of variance, with Bonferroni correction for multiple-comparison testing. For other metrics examined, see Tables 3 and 4. Estimated CPP is calculated as the difference between DBP and ventricular end-diastolic pressure (Nguyen et al., 2018). RPP is calculated as the product of heart rate and systolic blood pressure(Ansari et al., 2012). [Colour figure can be viewed at wileyonlinelibrary.com]

**Table 4. The magnitude of changes from baseline in cardiovascular function prior to and following AIH delivery across different experimental groups**

| Variables | Δ Post | | | Δ 90min | | |
|---|---|---|---|---|---|---|
| | AIH | TC | P | AIH | TC | P |
| **LV systolic function measures** | | | | | | |
| $P_{max}$ (mmHg) | $9 \pm 12$ | $2 \pm 3$ | 0.14 | $11 \pm 8$ | $2 \pm 5$ | **0.05** |
| $dP/dt_{max}$ (mmHg/s) | $897 \pm 1232$ | $99 \pm 376$ | 0.16 | $1046 \pm 975$ | $98 \pm 837$ | 0.08 |
| $P@dP/dt_{max}$ (mmHg) | $5 \pm 7$ | $0 \pm 2$ | 0.14 | $5 \pm 5$ | $0 \pm 3$ | 0.11 |
| CPP (mmHg) | $7 \pm 7$ | $-1 \pm 2$ | **0.009** | $6 \pm 6$ | $-1 \pm 4$ | **0.02** |
| RPP (mmHg·bpm) | $5746 \pm 6526$ | $-390 \pm 2195$ | **0.02** | $5042 \pm 4217$ | $-523 \pm 4728$ | **0.04** |
| **LV diastolic function measures** | | | | | | |
| $dP/dt_{min}$ (mmHg s$^{-1}$) | $-737 \pm 683$ | $60 \pm 287$ | **0.006** | $-749 \pm 615$ | $31 \pm 312$ | **0.007** |
| EDP (mmHg) | $-0.2 \pm 2.1$ | $0 \pm 0.5$ | $> 0.99$ | $-0.4 \pm 2$ | $-0.2 \pm 1$ | $> 0.99$ |
| **Systemic arterial pressure** | | | | | | |
| SBP (mmHg) | $9 \pm 12$ | $2 \pm 3$ | 0.13 | $11 \pm 8$ | $2 \pm 5$ | **0.05** |
| DBP (mmHg) | $7 \pm 8$ | $-1 \pm 3$ | **0.03** | $6 \pm 8$ | $-1 \pm 4$ | **0.05** |
| MAP (mmHg) | $8 \pm 9$ | $0 \pm 3$ | **0.05** | $8 \pm 7$ | $0 \pm 4$ | **0.05** |
| PP (mmHg) | $2 \pm 4$ | $3 \pm 1$ | $> 0.99$ | $5 \pm 5$ | $3 \pm 3$ | 0.45 |
| HR (bpm) | $15 \pm 24$ | $-9 \pm 13$ | **0.05** | $3 \pm 23$ | $-11 \pm 27$ | 0.42 |

*Note*: Data are presented as the mean ± SD. Statistical analysis was performed using a two-way (group × time) mixed analysis of variance, with Bonferroni correction for multiple-comparison testing. Estimated CPP is calculated as the difference between DBP and ventricular EDP (Nguyen et al., 2018). RPP is calculated as the product of HR and SBP(Ansari et al., 2012). The bolded p values are significant observations.

Abbreviations: AIH, hypoxia-exposed group ($n = 11$); DBP, diastolic blood pressure; $dP/dt_{max}$, the maximal rate of rise of the LV pressure; $dP/dt_{min}$, the maximal rate of decrease of the LV pressure; EDP, end-diastolic pressure; HR, heart rate; LV, left ventricle; MAP, mean arterial pressure; $P@dP/dt_{max}$, pressure at $dP/dt_{max}$, CPP, coronary perfusion pressure; $P_{max}$, maximal pressure of the LV pressure; PP, pulse pressure; RPP, rate pressure product; SBP, systolic blood pressure; TC, time control group ($n = 8$).

humans with chronic SCI) have utilized pharmacological blockades to isolate and determine the functional effects of parasympathetic control on cardiac (dys)function post-SCI. By blocking parasympathetic transmission to the heart using the muscarinic receptor antagonist atropine, we found no significant differences in the magnitude of changes (from baseline) for cardiac function (i.e. $dP/dt_{max}$, HR) and systemic blood pressure (i.e. DBP) between groups studied, and marked cardiovascular dysfunction remained in rats with SCI *vs.* Naive rats. The similar responses to atropine between SCI and Naive animals substantiate the presence of intact parasympathetic control of the heart at the same time as eliminating the potential role of this arm of the autonomic nervous system in cardiac dysfunction post-SCI (at least in an anaesthetized preparation and in a rat model of high-thoracic contusion SCI at 2 weeks post-injury), suggestive of shifting therapeutic focus towards restoring sympathetic control post-SCI.

To dissect any additional vascular influences on cardiac (dys)function, we added hexamethonium to the double blockade condition to further block autonomic transmission to the vasculature. In response to this additional hexamethonium, Naive animals showed a further significant reduction in cardiac function (i.e.

$dP/dt_{max}$) and systemic arterial pressure (i.e. DBP). Conversely, $dP/dt_{max}$ remained unchanged in SCI rats. Moreover, when we examined the association between the magnitude of reduction in DBP (i.e. representing the vascular influence) and $dP/dt_{max}$ from double to complete blockade condition, we found a lower (albeit still significant) regression slope for SCI *vs.* uninured (42 *vs.* 67) animals; that is, for every one unit drop of DBP, there is a greater reduction in $dP/dt_{max}$ in Naive *vs.* SCI rats. Although the heart–vascular interaction has previously been established in various clinical scenarios (Borlaug & Kass, 2008; Guinot et al., 2022; Latus et al., 2023; O'rourke, 1994), we are the first to utilize a comprehensive pharmacological manipulation approach to test this post-SCI. The fact that we found no major vascular effects on heart function post-SCI aligns with our prior publications, which have reported impairments in load independent indices of LV contractility post-SCI (Fossey et al., 2022; Hayes et al., 2021; Poormasjedi-Meibod et al., 2019; Squair et al., 2018; Williams et al., 2020).

Our findings from the pharmacological blockade aspect of this work, as well as suggestions from previous studies (Draghici & Taylor, 2018; Fossey et al., 2022; Lucci et al., 2021; Lujan et al., 2018; Zahner et al., 2011), clearly

confirm the key role of interrupted sympathetic control as a major contributor to reduced cardiac function and blood pressure post-SCI. As such, we subsequently examined whether we could neuromodulate the heart of SCI animals with AIH by re-activating the sublesional sympathetic circuits that control the heart and major blood vessels. Importantly, we and others have recently demonstrated that AIH induces sympathetic LTF in sublesional spinal circuits that can last up to 90 min following the stimulus in SCI rats (Ahmadian et al., 2025; Perim et al., 2023). Such AIH-induced augmentation in sympathetic nerve activity/blood pressure is also reported in able-bodied individuals (Edmunds et al., 2021; Jouett et al., 2017; Ott et al., 2020) and animals with intact spinal cords (Blackburn et al., 2018; Dick et al., 2007; Ostrowski et al., 2023).

In the present study, we assessed whether such AIH-induced sympathetic LTF can be utilized as a form of neuromodulation to increase the activity of the sympathetic pathways innervating the heart and vasculature and subsequently increase cardiac and systemic arterial pressure. We report for the first time that AIH is effective at neuromodulating the heart because AIH increased the pressure-generating capacity of the heart (e.g. $dP/dt_{max}$), systemic arterial pressure (e.g. DBP) and coronary perfusion 90 min following the cessation of AIH exposure. Our findings demonstrate the functional cardiovascular impact of AIH-induced sLTF (Ahmadian et al., 2025; Perim et al., 2023) and thereby introduce AIH as a potential neuromodulatory approach to improve resting cardiovascular function in people (and animals) living with SCI. To our knowledge, only a single recent study has reported on the blood pressure and HR response to AIH in individuals with SCI (Welch et al., 2024). In that study, no significant differences in blood pressure and a small increase in HR were reported following AIH exposure. Although the study by Welch et al. (2024) lacks any assessments of cardiac pressure generation capacity, stroke volume or systemic vascular resistance, the lack of increase in blood pressure post-AIH in their study could be a result of the more heterogeneous sample used (i.e. inclusion of individuals with injuries ranging from C3 to T6 and AIS grades A–D).

In considering the potential translational potential of AIH, we consider that our novel findings should be interpreted with caution. Although improving resting sympathetic tone would theoretically offset a number of key haemodynamic challenges that individuals with high-lesion SCI face (i.e. resting hypotension, orthostatic hypotension, risk of ischaemic cardiac events), it may exacerbate the severity of AD. AD describes a unique clinical scenario where sensory afferents activate intraspinal pathways that ultimately activate the unopposed sublesional sympathetic circuits to cause pronounced hypertension. Left untreated, AD has been associated with life-threatening hypertension and has been shown to increase the risk of myocardial infarction, stroke, and even death (Wan & Krassioukov, 2014). Thus, if the 'cost' of elevating sympathetic tone is that it would worsen the severity of AD, then this would need to be weighed carefully for individuals with SCI. On the other hand, the present study examined only the cardiovascular responses to a single exposure of AIH. It is entirely possible that if AIH was repeated on a daily basis then it may cause long-term plasticity within the sympathetic circuit and thus dampen the AD reflex altogether. Thus, future studies should examine the efficacy of daily AIH on long-term sympathetic plasticity and blood pressure regulation in the setting of SCI ahead of translation.

Although the findings from the present study are promising, a few considerations with respect to the choice of protocol need to be acknowledged. First, although we are confident our protocol elicits sympathetic LTF, it may not represent the most optimal 'cardiovascular' based AIH protocol. Thus, future studies investigating various doses/number of repeats/durations, as well as the potential impact of altering arterial carbon dioxide tension (i.e. combined hypo-/iso-/hypercapnic hypoxia), are welcomed. Second, we chose to use hyperoxia in the recovery between the bouts of hypoxia. This choice was designed to help limit the degree of hypotension experienced by the animals during each bout of hypoxia. Although hyperoxia may be expected to suppress the carotid body (and therefore offset the expected benefits of AIH), we have previously shown that an intact carotid body is not required for AIH-induced sympathetic LTF (Ahmadian et al., 2025). We therefore consider that the choice to use hyperoxia between bouts probably had no major impact on our study findings.

It should also be noted that we did find some between-group variability in AIH and TC groups, which manifested in our AIH group having significantly lower values for a number of cardiac indices at baseline. This group difference was probably caused either by subtle differences in the injury characteristics (force/displacement/position) from our contusion injury apparatus or, more probably, inherent within animal differences in the progression of the secondary injury. For example, if animals have a 'worse' secondary injury response (i.e. more inflammation and/or prolonged ischaemia) then it is expected they will have a more severe injury and present with reduced cardiovascular function. The magnitude of variation across our experimental groups is similar to that previously observed in humans with cervical SCI who are all classified as having a neurologically complete (AIS A) injury.

Furthermore, our sample included only male rats. Although our focus on male rats reflects the much larger proportion of males than females living with chronic SCI (National Spinal Cord Injury Statistical Centre,

2023), previous studies have shown sex differences in AIH-induced cardiovascular plasticity (Puri et al., 2021). Indeed, despite showing similar adjustments in muscle sympathetic activity post-AIH, only males had blood pressure above baseline post-AIH exposure (Jacob et al., 2020). Whether such sex differences in the blood pressure response to AIH are also present in the setting of SCI remains to be explored.

In conclusion, our findings confirm the well established notion that the lack of supraspinal sympathetic control is the main driver of cardiac (vascular) dysfunction following high-thoracic SCI. Our findings also uniquely demonstrate that AIH can neuromodulate the heart and cardiovascular system in a rat model of high-thoracic SCI. These observations identify adrenergic pathways as a focal point for therapeutic interventions, and additionally introduce AIH as an effective neuromodulatory tool for the cardiovascular system post-SCI, setting the stage for the chronic application of therapies leveraging AIH to rescue autonomic balance and cardiovascular function.

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

## Additional information

### Data availability statement

The data that support the findings of this study are available from the corresponding author upon reasonable request.

### Competing interests

The authors declare that they have no competing interests.

### Author contributions

C.R.W and M.A conceived and designed the study. M.A performed animal experiments. E.E conducted spinal cord injury procedures. M.A analyzed data. M.A, C.R.W and E.E wrote the manuscript. All authors reviewed the manuscript and approved the final version of the manuscript submitted for publication. All authors agree to be accountable for all aspects of the work. All persons designated as authors qualify for authorship, and all those who qualify for authorship are listed.

### Funding

This work was funded by International Spinal Research Trust & Christopher and Dana Reeve Foundation (Grant number: TRI006), Natu Sciences and Engineering Research Council of Canada (Grant number: RGPIN-2020-0624) and Blusson Integrated Cures Partnership (Grant number: GR016954).

### Acknowledgements

We thank all members of the Translational Integrative Physiology Laboratory who aided with the care of SCI animals.

### Keywords

animal model, autonomic nervous system, neuromodulation, parasympathetic, pharmacological blockade, spinal cord injury, sympathetic, therapeutic hypoxia

## Supporting information

Additional supporting information can be found online in the Supporting Information section at the end of the HTML view of the article. Supporting information files available:

**Peer Review History**

