## [Peer Review History · The Journal of Physiology]

Cardiac neuromodulation with acute intermittent hypoxia in rats with spinal cord injury

Mehdi Ahmadian, Erin Erskine, and Christopher R West

DOI: 10.1113/JP287676

Corresponding author(s): Christopher West (west@icord.org)

The following individual(s) involved in review of this submission have agreed to reveal their identity: Ken D O'Halloran (Referee #1)

Review Timeline:

Submission Date:	11-Sep-2024
Editorial Decision:	02-Jan-2025
Revision Received:	22-Jan-2025
Editorial Decision:	13-Feb-2025
Revision Received:	21-Feb-2025
Accepted:	26-Feb-2025

Senior Editor: Harold Schultz

Reviewing Editor: Diana Martinez

Transaction Report:

Dear Dr West,

Re: JP-RP-2024-287676 "Neuromodulation with acute intermittent hypoxia alleviates sympathetically-mediated cardiac dysfunction in rats with spinal cord injury" by Mehdi Ahmadian, Erin Erskine, and Christopher R West

Thank you for submitting your manuscript to The Journal of Physiology. It has been assessed by a Reviewing Editor and by 3 expert referees and we are pleased to tell you that it is acceptable for publication following satisfactory revision.

REVISION CHECKLIST:

We look forward to receiving your revised submission.

Yours sincerely,

Harold Schultz
Senior Editor
The Journal of Physiology

REQUIRED ITEMS

- Author photo and profile. First or joint first authors are asked to provide a short biography (no more than 100 words for one author or 150 words in total for joint first authors) and a portrait photograph. These should be uploaded and clearly labelled together in a Word document with the revised version of the manuscript. See Information for Authors for further details.

- You must start the Methods section with a paragraph headed Ethical approval (https://jp.msubmit.net/cgi-bin/main.plex?form_type=display_requirements#methods).

Research must comply with The Journal's policies regarding animal experiments (<https://physoc.onlinelibrary.wiley.com/hub/animal-experiments>) and adherence to these policies must be stated in the manuscript.

Authors should confirm in their Methods section that their experiments were carried out according to the guidelines laid down by their institution's animal welfare committee, including an ethics approval reference number. The Methods section must contain a statement about access to food, water and housing, details of the anaesthetic regime: anaesthetic used, dose and route of administration, and method of killing the experimental animals.

- Please upload separate high-quality figure files via the submission form.

- Please ensure that any tables are editable and in Word format, and wherever possible, embedded in the article file itself.

- Papers must comply with the Statistics Policy: https://jp.msubmit.net/cgi-bin/main.plex?form_type=display_requirements#statistics.

In summary:

- If $n \leq 30$, all data points must be plotted in the figure in a way that reveals their range and distribution. A bar graph with data points overlaid, a box and whisker plot or a violin plot (preferably with data points included) are acceptable formats.

- If $n > 30$, then the entire raw dataset must be made available either as supporting information, or hosted on a not-for-profit repository, e.g. FigShare, with access details provided in the manuscript.

- 'n' clearly defined (e.g. x cells from y slices in z animals) in the Methods. Authors should be mindful of pseudoreplication.

- All relevant 'n' values must be clearly stated in the main text, figures and tables.

- The most appropriate summary statistic (e.g. mean or median and standard deviation) must be used. Standard Error of the Mean (SEM) alone is not permitted.

- Exact p values must be stated. Authors must not use 'greater than' or 'less than'. Exact p values must be stated to three

significant figures even when 'no statistical significance' is claimed.

- Please include an Abstract Figure file, as well as the Figure Legend text within the main article file. The Abstract Figure is a piece of artwork designed to give readers an immediate understanding of the research and should summarise the main conclusions. If possible, the image should be easily 'readable' from left to right or top to bottom. It should show the physiological relevance of the manuscript so readers can assess the importance and content of its findings. Abstract Figures should not merely recapitulate other figures in the manuscript. Please try to keep the diagram as simple as possible and without superfluous information that may distract from the main conclusion(s). Abstract Figures must be provided by authors no later than the revised manuscript stage and should be uploaded as a separate file during online submission labelled as File Type 'Abstract Figure'. Please also ensure that you include the figure legend in the main article file. All Abstract Figures should be created using BioRender. Authors should use The Journal's premium BioRender account to export high-resolution images. Details on how to use and access the premium account are included as part of this email.

EDITOR COMMENTS

Reviewing Editor:

Ethics Concerns:

Question arose on the type of anesthesia used, the ethics editor has given their comments and found the explanation acceptable.

Comments to the Author (Required):

The authors are interested in determining whether interrupted bulbospinal sympathetic projections directly or indirectly contribute to cardiovascular dysfunction in spinal cord injury. The manuscript was reviewed by a reviewing editor (RE), an ethics editor, and two expert reviewers. The RE and reviewers acknowledge the potential impact this manuscript may have, but some items need to be addressed. One important point is the level at which results from different protocols are presented with respect to intermittent hypoxia. The authors need to clearly address whether a single exposure to IH is actually therapeutic as opposed to a metabolic adjustment.

Please also see 'Required Items' above.

Senior Editor:

Comments for Authors to ensure the paper complies with the Statistics Policy (Required):

The Journal asks that actual p values be shown throughout. Please include p values in Table 1 as in Tables 2-4.

Please include p values in Figures 1-4 as in Figure 5.

Comments to the Author:

Thank you for submission of your research article to the Journal of Physiology for consideration. The article has been reviewed by experts in the field and found to be potentially acceptable for publication pending adequate revision to address all of the concerns raised. Please address all comments from the external referees and reviewing editor as well as addressing the list of requirements or publication in the journal included in this letter and the link below.

<https://physoc.onlinelibrary.wiley.com/pb-assets/hub-assets/physoc/documents/TJP-Rigour-and-Reproducibility-Requirements-1724673661727.pdf>

REFeree COMMENTS

Referee #1:

Comments for Author (Required):

Thank you for submitting your manuscript to The Journal of Physiology. Some additional details pertaining to animal welfare are required.

The sequence of the narrative in the Methods section does not follow the sequential fate of the animals. It might be better to present the narrative with the description of anaesthesia induction, followed by surgical procedures and then specific details pertaining to measurements and protocol, finishing with method of killing. For example, as written, there is a need to include a description of surgical procedures as early as line 137.

Line 165: Please include details of the carrier gas.

Line 177. Please provide the route of administration.

Line 178-183: Thank you for providing a description of post-surgical care, revealing adherence to an excellent standard of care for the experimental animals.

Line 187. Please include details of the carrier gas.

Line 189: Although it may appear redundant, please re-state how adequacy of depth of anaesthesia was determined.

Line 195/6: What was the dose of urethane infusion?

Line 201: Please confirm that bolus injections were given when required against the backdrop of the constant urethane infusion.

Line 212: Thank you for the clear text provided in the description of the method of euthanasia. The Journal has recently adopted a policy change in that it no longer accepts the use of chloral hydrate for euthanasia given that more refined approaches are available. However, it is evident in your study that animals were deeply anaesthetised using urethane before administration of chloral hydrate. This is acceptable on the basis of the deep plane of anaesthesia established using urethane, which was confirmed in the study. As such, the approach does not contravene our recently published policy change (<https://physoc.onlinelibrary.wiley.com/doi/10.1113/JP286666>) .

Line 221. This section further describes in vivo measurements and is positioned after the description of method of euthanasia, therefore it requires description of induction of anaesthesia, assessment of adequacy of depth of anaesthesia, surgical procedures and method of euthanasia for these animals. It may therefore be better to move this section to earlier in the methods.

Referee #2:

Thank you for submitting your work to The Journal of Physiology. Below I have provided a point-by-point review of each section followed by a general review of each section and/or the overall paper. My goal is to provide feedback that I believe

will help improve the paper while maintaining the rigors of science and the Journal. I hope you find the review helpful, and thank you for your hard work.

Abstract:

No comments

Introduction:

Line 70: Cragg 2013 is a bit dated to stated the CV complications are "now one" of the leading causes of morbidity and mortality. Albeit true, new citations should be replace/added.

Line 97: It's difficult to state a single exposure to AIH is indeed therapeutic. Considering the authors' experience in exercise studies, if the outcome measure is blood pressure and one investigated exercise, would the rise in blood pressure following acute exercise be labelled as therapeutic?

Line 100: I don't understand the use of (i.e., key motor) here. It could be stylistic, but I would just assumed saying "show to improve key motor functions such as walking..."

Line 107: Yes, some use IH to elicit sympathetic function in acute studies, but as a therapy (ie repeated exposure) has shown to decrease blood pressure in humans with hypertension despite the presence of LTF of blood pressure on first and last days. I think this needs to be addressed considering the use of the word therapeutic in the introduction. Albeit recognizing impaired sympathetic function between individuals with SCI and hypertension are drastically different.

Overall introduction: I understand writing at a broader level, but much of the translational references come from the Vose paper and many animal studies that have varying protocols, especially the work in humans. What is presented for breathing plasticity, as a translatable component, has very distinct components (i.e. CO₂) that would need to be addressed. Ultimately, a wide breadth of protocols is covered that don't appear specific or precise to the protocol used in this study. Furthermore, the introduction does not support the use of 100% O₂ in the recovery bouts (see methods) and the potential impact of hyperoxia on the carotid bodies and sympathetic function. These are major components of protocols used to elicit plasticity of the various systems you addressed in the introduction. I believe refinement for the support of this IH protocol is required.

Methods:

Line 158: Again, using "treatment" remains difficult here

Line 239-243 and line 246-249: This protocol is not supported in your introduction and is drastically different from protocols presented in the Vose article. In fact, for the statements of IH and breathing, hyperoxia has shown to mitigate plasticity of breathing via the peripheral chemoreceptors. This should be addressed.

Line 262: Is this powered for within or group differences? Please state and justify if the study is powered for group or interaction effects.

Results:

Line 408: How does a single exposure alleviate cardiovascular dysfunction? This is in line with the ambiguity of a single exposure of IH being therapeutic.

Table 3: If all rats have the same SCI, how can you explain the differences in Pmax, SBP, PP, HR in the AIH group?

Please clarify the time of the post measurements. Were these immediately after hypoxia or 5 minutes after etc.

Discussion:

Line 460: I don't understand the use of "cardiac (vasculature)". Please clarify what is meant by this presentation

Line 461: Why is it neuromodulate now, but it was treatment/therapeutic and alleviate before?

Line 506-507: This is colloquial. This should simply state this aligns with previous research.

Line 518: There is also human evidence that supports this following hypoxia whether it be muscle sympathetic recordings or even blood pressure.

Line 525: Please specific the systems that are in agreement following these methods. As mentioned, hyperoxia has shown to mitigate vLTF in humans. Thus mixing protocols and species is impacting the precision of the text.

Line 527: Again, how is acute exposure restoring function? If this is interventional, and translatable, there are many studies that could be addressed here. Plus justify the use of a single exposure as therapeutic. If mentioning cardiovascular function in humans, it remains unclear why autonomic dysreflexia and orthostatic hypotension are not addressed. For example, wouldn't this augment autonomic dysreflexia?

Tables and Figures:

Table 1: No bolded p-values as written in the table legend

Tables: various number of digits expressed across measures. I would suggest standardizing to the same numbers and improving the formatting so all the values are on a single line, or for the decimals with many variables, keeping the {plus minus}SD on the second line so it is consistent.

Overall Impression:

Overall, this is a nicely written manuscript with very clear methods. The figures, especially the methodological component of the figures (1a, 2a etc) were appreciated and help improve the clarity of the data. Likewise, including the raw data with the magnitude of change strengthens the findings of this paper. My concerns are centered around the use of therapeutic and alleviation with a single exposure of hypoxia. I would contend a single exposure to IH is not therapeutic. Likewise, I have concerns regarding the introductions lack of support for this specific IH protocol; there is no mention of the impact of hyperoxia on the carotids which would have a significant impact on these findings and the protocol is used based on various studies for neural LTF, but only blood pressure and cardiac function were measured, which is not discussed. Additionally, the introduction mixes animal and human work which all have significant differences in the protocols employed. This is further complicated by the differences in IH protocols in humans that report changes in blood pressure (ie LTF of blood pressure, and how it changes with daily IH). Ultimately, I think the introduction lacks specificity for the specific IH protocol used.

Albeit not required, I believe the authors are missing the opportunity to discuss the potential translational component of their data (which is typically an option for resubmitted articles). Specifically, with SCI and sympathetic function, it would be beneficial to discuss how augmenting sympathetic function may be beneficial for orthostatic hypotension, but then autonomic dysreflexia could be discussed. This also opens the door to discuss daily IH exposure (ie interventional) and important blood pressure outcomes. Finally, I do believe a limitations section is required. It would clarify some of the questions addressed above, and it will provide the authors' an opportunity to discuss any potential order effects.

Referee #3:

The authors are interested in determining whether interrupted bulbospinal sympathetic projections directly or indirectly contribute to cardiovascular dysfunction in spinal cord injury. To determine this, the group used acute intermittent hypoxia, a well known model, can alter cardiac response post spinal cord injury.

This is a well-written manuscript, however, I do have some points that need to be addressed. There are the usage of dated citations within the introduction, recent citations should be added. Additionally, the introduction falls short of convincing the reader that this is the best model to use for this. Clarity of human and animal results are needed to present the issue at hand in the introduction.

Some of the data is unclear, for example if all of the rats are exposed to the same SCI, how are there differences in baseline? Overall, it is unclear with the data presented how a single bout of AIH can restore function in SCI. This should be clarified.

END OF COMMENTS

POINT-BY-POINT RESPONSES TO EDITORS' AND REVIEWERS'
COMMENTS

**“Neuromodulation with acute intermittent hypoxia alleviates
sympathetically-mediated cardiac dysfunction in rats with
spinal cord injury”, ID: JP-RP-2024-287676**

Ahmadian et al

EDITOR COMMENTS

Reviewing Editor:

Ethics Concerns:

Question arose on the type of anesthesia used, the ethics editor has given their comments and found the explanation acceptable.

Comments to the Author (Required):

The authors are interested in determining whether interrupted bulbospinal sympathetic projections directly or indirectly contribute to cardiovascular dysfunction in spinal cord injury. The manuscript was reviewed by a reviewing editor (RE), an ethics editor, and two expert reviewers. The RE and reviewers acknowledge the potential impact this manuscript may have, but some items need to be addressed. One important point is the level at which results from different protocol are presented with respect to intermittent hypoxia. The authors need to clearly address whether a single exposure to IH is actually therapeutic as opposed to a metabolic adjustments.

Please also see 'Required Items' above.

Author's Response: Thank you for your time spent reviewing our work. We tried to address these concerns in the revised manuscript.

Senior Editor:

Comments for Authors to ensure the paper complies with the Statistics Policy (Required):

1. The Journal asks that actual p values be shown throughout. Please include p values in Table 1 as in Tables 2-4.
2. Please include p values in Figures 1-4 as in Figure 5.

Authors' Response: Thank you for these comments. These all have been addressed in the revised manuscript.

Comments to the Author:

3. Thank you for submission of your research article to the Journal of Physiology for consideration. The article has been reviewed by experts in the field and found to be potentially acceptable for publication pending adequate revision to address all of the concerns raised. Please address all comments from the external referees and reviewing editor as well as addressing the list of requirements or publication in the journal included in this letter and the link below.

<https://physoc.onlinelibrary.wiley.com/pb-assets/hub-assets/physoc/documents/TJP-Rigour-and-Reproducibility-Requirements-1724673661727.pdf>

Authors' Response: We thank the editor for the opportunity to reconsider our manuscript following a satisfactory revision. We have tried to fully address the suggestions made by editors and reviewers in the revised manuscript as well as adhere to the requirements mentioned in the decision letter for submitting a revision to the journal.

REFEREE COMMENTS

Referee #1:

Thank you for your time spent reviewing our work and for your constructive comments.

Comments for Author (Required):

1. Thank you for submitting your manuscript to The Journal of Physiology. Some additional details pertaining to animal welfare are required.

The sequence of the narrative in the Methods section does not follow the sequential fate of the animals. It might be better to present the narrative with the description of anaesthesia induction, followed by surgical procedures and then specific details pertaining to measurements and protocol, finishing with method of killing. For example, as written, there is a need to include a description of surgical procedures as early as line 137.

Authors' Response: Thank you for this comment. We agree that the method of killing (i.e., euthanasia) is better suited for the end of methods, and we moved this section accordingly in the revised manuscript. We prefer having an "Experimental Design" section at the beginning of the methods immediately after the "Ethical Approval" section to help readers better understand the overall design of the method for each experiment. The rest, in our opinion, now follows the sequential fate of the animals; that is, SCI injury and recovery, terminal preparations, AIH exposure, and euthanasia.

2. Line 165: Please include details of the carrier gas.

Authors' Response: Thank you for this comment. This has been addressed in the revised manuscript.

3. Line 177. Please provide the route of administration.

Authors' Response: Thank you for this comment. The route of administration for lactated Ringer's solution was subcutaneous and is mentioned in the text.

4. Line 178-183: Thank you for providing a description of post-surgical care, revealing adherence to an excellent standard of care for the experimental animals.

Authors' Response: Thank you!

5. Line 187. Please include details of the carrier gas.

Authors' Response: Thank you for this comment. This has been addressed in the revised manuscript.

6. Line 189: Although it may appear redundant, please re-state how adequacy of depth of anaesthesia was determined.

Authors' Response: Thank you for this comment. The requested information has been added, and it now reads as below.

“Once a surgical plan of anesthesia was reached using isoflurane (i.e., no hind-paw [for Naive animals] and front-paw [for SCI animals] withdrawal/corneal reflexes [for both Naive and SCI animals]),”

7. Line 195/6: What was the dose of urethane infusion?

Authors' Response: Thank you for this comment. The dose of urethane infusion and time to transfer to urethane has been added and now reads as below.

“Urethane was then delivered IV as a constant-rate infusion (6 mL/h) over ~20–25 min (i.e., up to a final dose of 2.1 g/kg)”

8. Line 201: Please confirm that bolus injections were given when required against the backdrop of the constant urethane infusion.

We confirm that we topped up the urethane levels throughout the experiment if required (0.1 g/kg, as many times as required - typically not more than two).

9. Line 212: Thank you for the clear text provided in the description of the method of euthanasia. The Journal has recently adopted a policy change in that it no longer accepts the use of chloral hydrate for euthanasia given that more refined approaches are available. However, it is evident in your study that animals were deeply anaesthetised using urethane before administration of chloral hydrate. This is acceptable on the basis of the deep plane of anaesthesia established using urethane, which was confirmed in the study. As such, the approach does not contravene our recently published policy change (<https://physoc.onlinelibrary.wiley.com/doi/10.1113/JP286666>) .

Authors' Response: Thank you! Please also note that we are now no longer using chloral hydrate in our current and future studies.

10. Line 221. This section further describes in vivo measurements and is positioned after the description of method of euthanasia, therefore it requires description of induction of anaesthesia, assessment of adequacy of depth of anaesthesia, surgical procedures and method of euthanasia for these animals. It may therefore be better to move this section to earlier in the methods.

Authors' Response: Thank you for this comment. Please see our response to your comment #1.

Referee #2:

1. Thank you for submitting your work to The Journal of Physiology. Below I have provided a point-by-point review of each section followed by a general review of each section and/or the overall paper. My goal is to provide feedback that I believe will help improve the paper while maintaining the rigors of science and the Journal. I hope you find the review helpful, and thank you for your hard work.

Author's Response: Thank you for your time spent reviewing our work and for your constructive comments.

Abstract:

No comments

Introduction:

2. Line 70: Cragg 2013 is a bit dated to stated the CV complications are "now one" of the leading causes of morbidity and mortality. Albeit true, new citations should be replace/added.

Authors' Response: Thank you for this comment. Recent citations have been added to address this comment.

3. Line 97: It's difficult to state a single exposure to AIH is indeed therapeutic. Considering the authors' experience in exercise studies, if the outcome measure is blood pressure and one investigated exercise, would the rise in blood pressure following acute exercise be labelled as therapeutic?

Authors' Response: Thank you for this comment. We appreciate the point being raised and agree that the use of the term 'therapeutic' is incorrect here. In the revised manuscript, we now refer to AIH as a "neuromodulatory" intervention throughout the text. This term also is more appropriate for the topic of the special issue for which this paper is being considered.

4. Line 100: I don't understand the use of (i.e., key motor) here. It could be stylistic, but I would just assumed saying "show to improve key motor functions such as walking..."

Authors' Response: Thank you for this comment. We have modified the text for better clarity, and it now reads as below.

"AIH has previously been shown to effectively improve key motor functions such as walking, hand/arm movement, and breathing in humans and animals with SCI (Vose et al., 2022)."

5. Line 107: Yes, some use IH to elicit sympathetic function in acute studies, but as a therapy (ie repeated exposure) has shown to decrease blood pressure in humans with hypertension despite the presence of LTF of blood pressure on first and last days. I think this needs to be addressed considering the use of the word therapeutic in the introduction. Albeit recognizing impaired sympathetic function between individuals with SCI and hypertension are drastically different.

Authors' Response: Thank you for this comment. Please see our response to comment #3 regarding the word therapeutic. As you pointed out, we performed only a single exposure. Whilst we believe a single exposure may in fact be immediately therapeutic (i.e., if someone has devastating resting hypotension and this can be alleviated with a single-bout of AIH then we would argue that the AIH was in fact therapeutic) we have not studied the effect of repeated exposures (i.e., using AIH as a daily therapy). In keeping with how the term "therapeutic" is more widely understood in the field we have removed this term from the manuscript.

6. Overall introduction: I understand writing at a broader level, but much of the translational references come from the Vose paper and many animal studies that have varying protocols, especially the work in humans. What is presented for breathing plasticity, as a translatable component, has very distinct components (i.e. CO₂) that would need to be addressed. Ultimately, a wide breadth of protocols is covered that don't appear specific or precise to the protocol used in this study. Furthermore, the introduction does not support the use of 100% O₂ in the recovery bouts (see methods) and the potential impact of hyperoxia on the carotid bodies and sympathetic function. These are major components of protocols used to elicit plasticity of the various systems you addressed in the introduction. I believe refinement for the support of this IH protocol is required.

Authors' Response: Thank you for this comment. Whilst we absolutely agree that the wide breadth of protocols makes interpretation of findings across studies hard to reconcile, the purpose of our introduction

was to introduce the concept that AIH can be used to neuromodulate various functions post-SCI, of which ventilatory function and breathing plasticity are the most studied. This is akin to the field of spinal stimulation, where many different levels of the spinal cord are stimulated with a view to improving function in various systems (i.e., respiratory/cardiovascular/bladder). In our previous version we wanted to do justice to the previous literature on this topic and acknowledge that we are absolutely not the first to investigate AIH in the field of SCI, but rather we are the first to examine whether AIH can be used to neuromodulate the heart. Since our paper did not aim to determine the 'optimum dose' of AIH to improve cardiovascular function we believe that an exhaustive review of the various protocols/doses used to date would overly complicate the introduction to this work. Nevertheless, we agree that highlighting our protocol is different to that used in other systems is important and have added this note to the introduction. We have also refined the rationale for the present protocol by citing our now published paper that examines mechanisms of AIH-induced sympathetic long-term facilitation (sLTF) in rodents with SCI (see also response to comments #8, 22 and 23). In line with your additional comments, we have also added a 'limitations' section to the manuscript where we address the role of hyperoxia between bouts. We hope you find the new introduction more appropriate to setup the current study rationale. For ease of review, we have pasted the amended paragraph from the introduction below:

"Neuromodulatory interventions are gaining widespread traction within the field of SCI. One of the main neuromodulatory options currently employed is acute intermittent hypoxia (AIH; episodic exposure of humans/animals to brief periods of low inspired oxygen concentration). AIH has previously been shown to effectively improve key motor functions such as walking, hand/arm movement, and breathing in humans and animals with SCI (Vose et al., 2022). These AIH-induced beneficial effects have been attributed to neural plasticity manifesting as long-term facilitation (LTF; a long-lasting increase in motor nerve activity post-AIH), which for respiratory neural plasticity (i.e., phrenic LTF) can manifest functionally as ventilatory LTF (vLTF: a long-lasting increase in minute ventilation post-AIH) in humans and awake animals (Mitchell & Johnson, 2003; Griffin et al., 2012; Vermeulen et al., 2020; Panza et al., 2023). Recently, we, and others, have documented that AIH also evokes sympathetic neuroplasticity (i.e., sympathetic LTF) in spinally injured animals – albeit using different injury models (contusion vs. hemisection) and AIH protocols (10 × 1 min vs. 3 × 5 min) (Perim et al., 2023; Ahmadian et al., 2025). The presence of AIH-induced sympathetic LTF post-SCI raises the intriguing possibility that AIH may yield functional benefits by neuromodulating the heart post-SCI.

Here, using sequential pharmacological blockade of autonomic transmission to the heart and vasculature in a rodent model of high-thoracic SCI, we first examined whether heart dysfunction following SCI is a consequence of either weakened direct or indirect sympathetic control, altered parasympathetic control, or a combination of both. We subsequently examined whether one session of AIH applied two weeks following injury neuromodulates the heart in our rodent model of SCI. To address these aims we used a contusion SCI model (i.e., leaving some neural pathways intact post-injury) to better mimic clinical scenario of SCI (~ 70% incomplete, National Spinal Cord Injury Statistical Center, 2023). It was hypothesized that 1) altered (impaired) direct and indirect sympathetic control, not parasympathetic control, underlies impaired heart function post-SCI, and 2) one session of AIH delivery neuromodulate the heart post-SCI."

Methods:

7. Line 158: Again, using "treatment" remains difficult here

Authors' Response: Thank you for this comment. Please see our response to your comment #3.

8. Line 239-243 and line 246-249: This protocol is not supported in your introduction and is drastically different from protocols presented in the Vose article. In fact, for the statements of IH and breathing, hyperoxia has shown to mitigate plasticity of breathing via the peripheral chemoreceptors. This should be addressed.

Authors' Response: Thank you for this comment. We understand the concern being raised regarding the protocol of the current study. In the time that this paper has been under review, we have published an additional paper examining the mechanisms of sLTF in response to AIH in rodents with SCI (Ahmadian et

al., 2025). Whilst this additional paper has very distinct aims from the present paper (CV responses to AIH) we believe the data collected in that paper help address the “protocol” concern. In brief (all included in the additional publication), when we began investigating the cardiovascular effects of AIH in SCI, we employed the classic 3 x 5 min AIH protocol commonly used to study phrenic LTF in rodents with SCI. However, when we tested this protocol in four SCI animals, we found that none of the animals could survive the AIH protocol due to their inability to maintain BP. We concluded and attributed these observations to the more severe midline contusion model used in our study vs. the hemi-contusion model used by labs studying phrenic LTF. To help identify a more appropriate protocol to study the cardiovascular responses to AIH, we explored the AIH protocol(s) commonly used in studies examining sympathetic neuroplasticity in intact animals (i.e., 10 x 45s exposures; for original work, see Dick et al. 2007, Exp. Physiol. 92:87-97). In intact animals, AIH is typically used as a model for sleep apnea (repetitive short-duration exposures to hypoxia). Because it is still not entirely clear where the line between ‘beneficial’ and ‘detrimental’ intermittent hypoxia may lie (Please see Navarrete-Opazo and Mitchel 2014), out of abundance of caution we lengthened the exposure to 60s (i.e., to avoid the rapid alterations in the level of hypoxia that is characteristic of sleep apnea). We have subsequently utilized the 10x60s exposure in all of our AIH studies and recently demonstrated that this protocol elicits robust sLTF in rodents with the identical SCI to that in the present study. In our other publication we also demonstrated that sLTF is present even after carotid-body denervation prior to AIH exposure, insinuating that the mechanisms of AIH-induced phrenic LTF (and the role of hyperoxia) differ from that of AIH-induced sLTF.

In order not to overly complicate the methods section of the present manuscript, and in light of our recent publication demonstrating this protocol elicits robust sLTF, we have chosen to cite this paper in the methods and refrain from a lengthy discussion on protocol optimization. In reality, we still do not know whether the current protocol is ‘optimal’ as there are a dizzying number of combinations of doses/durations/exposures that could be tested. We do, however, have the benefit of now being able to demonstrate that this protocol does elicit sLTF in our model of SCI. We have now modified the “AIH protocol” section to cite our other paper on this topic and additionally included a small limitation section to state that we still do not understand whether the current protocol is the ‘ideal’ or whether, for example, a hypercapnic-hypoxia protocol may be further beneficial as is the case in the respiratory circuits.

*“For hypoxia delivery, we used commercially available small rodent ventilation (VentElite, Harvard Apparatus, Holliston, MA, USA) and a computer-controlled gas mixer (CWE, Ardmore, PA) to expose animals to 10 bouts of %10 O₂ (balanced with nitrogen) lasting for 1 min interspersed with 2 mins of breathing %100 O₂. **This protocol is a modified iteration of the hypoxia protocol that has previously shown to effectively elicit sympathetic LTF in intact rodents (Dick et al., 2007) and is the identical protocol that we used in our recent study demonstrating AIH induces robust sLTF in rodents with SCI (Ahmadian et al., 2025).**”*

10. Line 262: Is this powered for within or group differences? Please state and justify if the study is powered for group or interaction effects.

Authors’ Response: Thank you for this comment. It is powered for within-group differences as SCI animals present with lower resting cardiovascular function, rendering detection of within-group (pre vs. drug trial) significant differences hard. This power analysis gave us confidence that the number of animals included was sufficient to detect any significant changes during drug trials (vs. baseline) in our SCI animals with reduced resting cardiovascular function.

Results:

11. Line 408: How does a single exposure alleviate cardiovascular dysfunction? This is in line with the ambiguity of a single exposure of IH being therapeutic.

Authors’ Response: Thank you for this comment. Please see our response to your comment #5.

12. Table 3: If all rats have the same SCI, how can you explain the differences in Pmax, SBP, PP, HR in the AIH group?

Authors' Response: Thank you for this comment. We performed our randomization at the time of SCI before we knew what the 'resting cardiovascular function' of the rats were (as we only knew this from the terminal preparation). Our injury model uses a weight-drop impactor device to 'contuse' the spinal cord. As such, subtle differences in the injury characteristics (force/displacement/position), or more likely inherent within animal differences in the progression of the secondary injury likely explain why we have such inter-individual differences in function. If animals have a 'worse' secondary injury response (i.e., more inflammation and/or prolonged ischemia) then it is expected they will have a more severe injury. The magnitude of variation is also similar to that observed even within humans with cervical SCI who are all classified as having a neurologically complete (AIS A) injury. Thus, we think this variation makes our study all the more relevant to the human condition. In essence, we feel that we just got unlucky that these baseline differences between the groups exist, but attribute this to the inherent variability in the model and our timing of randomization (i.e., the randomization before measuring baseline CV function).

13. Please clarify the time of the post measurements. Were these immediately after hypoxia or 5 minutes after etc.

Authors' Response: Thank you for this comment. The time of the post-measurements has been clarified, and it now reads as below:

"To explore whether AIH is able to neuromodulate the heart post-SCI, two groups of chronic SCI rats (AIH and TC) were instrumented with cardiac and arterial catheters and assessed for cardiovascular function at baseline and different time points (immediately when the final hypoxic bout was over [1 min] and 90 min) post-exposure."

Discussion:

14. Line 460: I don't understand the use of "cardiac (vasculature)". Please clarify what is meant by this presentation.

Authors' Response: Thank you for this comment. The word "vascular" has now been eliminated.

15. Line 461: Why is it neuromodulate now, but it was treatment/therapeutic and alleviate before?

Authors' Response: Thank you for this comment. We have now used the term "neuromodulate" throughout the text.

16. Line 506-507: This is colloquial. This should simply state this aligns with previous research.

Authors' Response: Thank you for this comment. This has been addressed in the revised manuscript.

17. Line 518: There is also human evidence that supports this following hypoxia whether it be muscle sympathetic recordings or even blood pressure.

Authors' Response: Thank you for this comment. To our knowledge, there is no study in individuals with SCIs documenting sLTF or blood pressure elevation post-AIH. We assume your point is directed toward able-bodied individuals (and animals), and we have now modified the text to accommodate your suggestion, and it reads as below:

"Importantly, we, and others, have recently demonstrated that AIH induces sympathetic LTF in sub-lesional spinal circuits that can last up to 90 min following the stimulus in SCI rats (Ahmadian et al., 2025; Perim et al., 2023). Such AIH-induced augmentation in sympathetic nerve activity/blood pressure is also reported in able-bodied individuals (Lusina et al., 2006; Ott et al., 2020; Edmunds et al., 2021) and

animals with intact spinal cords (Dick et al., 2007; Blackburn et al., 2018; Maruyama et al., 2019; Ostrowski et al., 2023)."

18. Line 525: Please specify the systems that are in agreement following these methods. As mentioned, hyperoxia has shown to mitigate vLTF in humans. Thus mixing protocols and species is impacting the precision of the text.

Authors' Response: Thank you for this comment. We tried to modify the text to address this concern, and it now reads as below:

"Our findings demonstrate the functional cardiovascular impact of AIH-induced sLTF (Perim et al., 2023; Ahmadian et al., 2025), and thereby introduce AIH as a potential neuromodulatory means to improve resting cardiovascular function in people (and animals) living with SCI."

19. Line 527: Again, how is acute exposure restoring function? If this is interventional, and translatable, there are many studies that could be addressed here. Plus justify the use of a single exposure as therapeutic. If mentioning cardiovascular function in humans, it remains unclear why autonomic dysreflexia and orthostatic hypotension are not addressed. For example, wouldn't this augment autonomic dysreflexia?

Authors' Response: Thank you for this comment. We believe the answers to these questions are provided in our responses to your comments #8 and #22-23.

Tables and Figures:

20. Table 1: No bolded p-values as written in the table legend

Authors' Response: Thank you for pointing this out. This has been addressed in the revised manuscript.

21. Tables: various number of digits expressed across measures. I would suggest standardizing to the \rs and improving the formatting so all the values are on a single line, or for the decimals with many variables, keeping the {plus minus}SD on the second line so it is consistent.

Authors' Response: Thank you for this comment. We have considered your comment in the revised manuscript and numbers are now consistent in tables.

Overall Impression:

22. Overall, this is a nicely written manuscript with very clear methods. The figures, especially the methodological component of the figures (1a, 2a etc) were appreciated and help improve the clarity of the data. Likewise, including the raw data with the magnitude of change strengthens the findings of this paper. My concerns are centered around the use of therapeutic and alleviation with a single exposure of hypoxia. I would contend a single exposure to IH is not therapeutic. Likewise, I have concerns regarding the introduction's lack of support for this specific IH protocol; there is no mention of the impact of hyperoxia on the carotids which would have a significant impact on these findings and the protocol is used based on various studies for neural LTF, but only blood pressure and cardiac function were measured, which is not discussed. Additionally, the introduction mixes animal and human work which all have significant differences in the protocols employed. This is further complicated by the differences in IH protocols in humans that report changes in blood pressure (ie LTF of blood pressure, and how it changes with daily IH). Ultimately, I think the introduction lacks specificity for the specific IH protocol used.

23. Albeit not required, I believe the authors are missing the opportunity to discuss the potential translational component of their data (which is typically an option for resubmitted articles). Specifically, with SCI and sympathetic function, it would be beneficial to discuss how augmenting sympathetic function may be beneficial for orthostatic hypotension, but then autonomic

dysreflexia could be discussed. This also opens the door to discuss daily IH exposure (ie interventional) and important blood pressure outcomes. Finally, I do believe a limitations section is required. It would clarify some of the questions addressed above, and it will provide the authors' an opportunity to discuss any potential order effects.

Authors' Response: Thank you for these comments and your positive assessment of our work. Given the responses in this document and the fact that we did not seek to 'identify an optimum protocol" we would prefer to steer clear of an exhaustive discussion of the pros/cons/differences in protocols across animal, humans and systems. Now our prior paper has been published, we would prefer to keep the introduction more broad and cite our paper to demonstrate that our AIH protocol does elicit sLTF post-SCI. We do, however, agree that adding both a limitations section and translational potential section would be welcome additions and have taken the opportunity to add these sections at the end of the discussion to highlight the potential impact of our findings in clinical scenarios. We have updated the manuscript accordingly and copied these new sections below for ease of reference:

"In considering the potential translational potential of AIH, we believe that our novel findings should be interpreted with caution. Whilst improving resting sympathetic tone would theoretically offset a number of key hemodynamic challenges that individuals with high-lesion SCI face (i.e., resting hypotension, orthostatic hypotension, risk of ischemic cardiac events) it may exacerbate the severity of autonomic dysreflexia (AD). AD describes a unique clinical scenario where sensory afferents activate intraspinal pathways that ultimately activate the unopposed sub-lesional sympathetic circuits to cause pronounced hypertension. Left-untreated, AD has been associated with life-threatening hypertension and has been shown to increase the risk of myocardial infarction, stroke, and even death (Wan & Krassioukov, 2014). Thus, if the 'cost' of elevating sympathetic tone is that it would worsen the severity of AD, then this would need to be weighed carefully for individuals with SCI. On the other hand, our study examined only the cardiovascular responses to a single exposure of AIH. It is entirely possible that if AIH was repeated on a daily basis then it may cause long-term plasticity within the sympathetic circuit and thus dampen the AD reflex altogether. Thus, future studies should examine the efficacy of daily AIH on long-term sympathetic plasticity and blood pressure regulation in the setting of SCI ahead of translation.

Whilst findings from the present work are promising, a few considerations with respect to the choice of protocol need to be acknowledged. Firstly, although we are confident our protocol elicits sympathetic LTF it may not represent the most optimal "cardiovascular" based AIH protocol. Thus, future studies investigating various doses/number of repeats/durations, as well as the potential impact of altering arterial carbon dioxide tension (i.e., combined hypo-/iso-/hypercapnic hypoxia), are welcomed. Second, we chose to use hyperoxia in the recovery between the bouts of hypoxia. This choice was designed to help limit the degree of hypotension experienced by the animals during each bout of hypoxia. Whilst hyperoxia may be expected to suppress the carotid body (and therefore offset the expected benefits of AIH), we have previously shown that an intact carotid body is not required for AIH-induced sympathetic LTF (Ahmadian et al., 2025). We believe, therefore, that the choice to use hyperoxia between bouts was unlikely to have majorly impacted our study findings.

In conclusion, our findings confirm the well-established notion that the lack of supraspinal sympathetic control is the main driver of cardiac (vascular) dysfunction following high-thoracic SCI. Our findings also uniquely demonstrate that AIH can neuromodulate the heart and cardiovascular system in a rat model of high-thoracic SCI. These observations identify adrenergic pathways as a focal point for intervention while additionally introducing AIH as an effective neuromodulatory tool for the cardiovascular system post-SCI, setting the stage for chronic application of therapies leveraging AIH to rescue autonomic balance and cardiovascular function."

Referee #3:

1. The authors are interested in determining whether interrupted bulbospinal sympathetic projections directly or indirectly contribute to cardiovascular dysfunction in spinal cord injury. To determine this, the group used acute intermittent hypoxia, a well known model, can alter cardiac response post spinal cord injury.

2. This is a well-written manuscript, however, I do have some points that need to be addressed. There are the usage of dated citations within the introduction, recent citations should be added. Additionally, the introduction falls short of convincing the reader that this is the best model to use for this. Clarity of human and animal results are needed to present the issue at hand in the introduction.

Author's Response: Thank you for your time spent reviewing our work and for your constructive comments. Citations have been updated to include more recent citations. Your comment is largely in line with reviewer #2 and we have modified the introduction accordingly (please see our response to reviewer 2's comment #6).

3. Some of the data is unclear, for example if all of the rats are exposed to the same SCI, how are there differences in baseline? Overall, it is unclear with the data presented how a single bout of AIH can restore function in SCI. This should be clarified.

Authors' Response: Thank you for this comment. Your valid concerns are again aligned with Reviewer 2's concerns and have been addressed in the revised manuscript. Please see our responses to Reviewer 2's comments #5 and #12.

Dear Chris

I'm just emailing to enquire whether you might be able to revise your article within the next few days?

If you can get your revised version to us by the middle of next week (Wednesday 26 Feb), we will certainly be able to include it in our 'Cardiac Neurobiology: Concepts to Clinic' special issue (if accepted).

We are almost at the point of closing this special issue off so that we can finalise publication.

If you need more time, that is fine - but we will no longer be able to guarantee inclusion of your article within the special issue. It may need to be a standalone item.

Hoping you can revise in the next few days...

Many thanks!

Diana

JP

Dear Dr West,

Re: JP-RP-2025-287676R1 "Cardiac neuromodulation with acute intermittent hypoxia in rats with spinal cord injury" by Mehdi Ahmadian, Erin Erskine, and Christopher R West

Thank you for submitting your manuscript to The Journal of Physiology. It has been assessed by a Reviewing Editor and by 3 expert referees and we are pleased to tell you that it is acceptable for publication following satisfactory revision.

Your revised manuscript should be submitted online using the link in your Author Tasks <https://jp.msubmit.net/cgi-bin/main.plex>

REVISION CHECKLIST:

Check that your Methods section conforms to journal policy: <https://jp.msubmit.net/cgi-bin/main.plex>

Check that data presented conforms to the statistics policy: <https://jp.msubmit.net/cgi-bin/main.plex>

- 'Potential Cover Art' for consideration as the issue's cover image
- Appropriate Supporting Information (Video, audio or data set: see <https://jp.msubmit.net/cgi-bin/main.plex>)

We look forward to receiving your revised submission.

Yours sincerely,

Harold Schultz
Senior Editor
The Journal of Physiology

EDITOR COMMENTS

Reviewing Editor:

The manuscript and revisions were reviewed by two reviewers and the reviewing editor. The authors have been responsive to the comments, however, some questions remain from reviewer 2. Specifically, reviewer 2 comments on the lack of explanation on the difference of baselines, the sex difference, and the weak citations concerning translational aspects.

Senior Editor:

Thank you for submission of your revised research article to the Journal of Physiology for consideration. Referee 2 has a few additional concerns raised which need to be addressed. Please address all comments from the external referee as well as addressing the list of requirements or publication in the journal.

Please also note that the Data Availability statement requires revision. Data in the manuscript is summary data only. The journal requires authors to permit access to all data incurred in the study if requested in a reasonable fashion and as allowed by any restrictions. (see the Journal's Rigour and Reproducibility document).

We suggest the following wording: The data that support the findings of this study are available from the corresponding author upon reasonable request.

<https://physoc.onlinelibrary.wiley.com/pb-assets/hub-assets/physoc/documents/TJP-Rigour-and-Reproducibility-Requirements-1724673661727.pdf>

REFeree COMMENTS

Referee #1:

Thank you for carefully addressing the previous points raised and for making changes to the text. All issues have been

satisfactorily addressed.

Referee #2:

Thank you for resubmitting your work to The Journal of Physiology. Below I have provided commentary on specific components addressed by the authors. I hope you find the review helpful, and thank you for your hard work.

Abstract:

No comments

Introduction:

We agree that there is an opportunity for single exposure of IH to be therapeutic, but that is not cohesive with the project. The new terminology helps streamline the paper for clarity.

Stylistically I still think the introduction does not focus in on the relevance of the protocol used in this study, but this is a point I can concede considering the other paper from the same laboratory is now published and presented. No further concerns

Methods:

Results:

Unsure if you want to discuss this in the results or discussion, but I still contend the variability in baseline function should be directly addressed. I do not think it takes away from the paper, but it is difficult to interpret a paper fully when statistical differences are found, but not addressed. This by no means requires extensive discussion, but I do believe it should be addressed.

Discussion:

Line 523: Ott and Edmunds are investigating sex differences which is not exactly the sexes in this investigation. This should be addressed in the limitations. Likewise, despite the title, Lusina did not complete intermittent hypoxia. The protocol used in that paper was continuous hypoxia over several days, which the differences from day to day is what they deemed "intermittent" (a brief discussion on issues with terminology is available PMID: 37560767 if clarification is needed on this statement). This paper showed increases in MSNA without increases in BP. Thus, I am unsure this is the correct citation for human work. In addition to Ott and Edmunds, acute exposure and alterations in BP in humans could include (ones not cited, if you cited them I did not include them): PMIDs:

1) LTF of SBP: 35721527

2) LTF of MAP or DBP: 28082332, 31805605

3) LTF of MSNA (albeit they used voluntary apneas with hypoxia: 15242836, 14555683

4) LTF of HRV: 28119623, 18403450

5) Inconsistencies of BP post AIH because of sex: 32966122

6) An in-depth review on human evidence, both within and immediately following AIH is available and may also be suitable for citation. This review covers blood pressure as well as HR: 33781731

Please note these are not required to be cited, but I would consider them better citations to add instead of Lusina. Likewise, these would facilitate any discussion of sex differences for human data. I'm sure there may be a few newer papers, but I am less familiar. I know Univ Florida did publish a manuscript on CV outcomes in SCI (albeit a lot of missing data) that showed no LTF in breathing or BP, that may be worth discussing.

The new text is undercited. This all cannot be a thesis statement. Repeated exposure in those with hypertension, albeit not SCI but still autonomic dysfunction, supports the reduction in BP as well as the BP response (peak) to hypoxia in humans. PMID: 35015980, 35721527 and those after MI: 15262041

Likewise, there are many editorials addressing doses/numbers etc that could be cited here.

I believe the impact, or potential impact, of sex still needs to be addressed considering the translatable papers cited and your rat population. This does not have to be extensive, but it is important to address.

Tables and Figures:

Overall Impression:

Overall, I believe the author's addressed most of my concerns adequately, but there some of the new components are undercited considering there are publications that directly address these concerns. The citations for translatable work are somewhat weak and I believe there are better citations- some examples were provided. Please note, none of this, in my opinion, requires extensive citations or discussion, but an easy opportunity to improve precision of the translatability of your work to humans.

Referee #3:

Thank you to the authors for responding to my comments. I believe they have adequately addressed my concerns.

END OF COMMENTS

POINT-BY-POINT RESPONSES TO EDITORS' AND REVIEWERS' COMMENTS

"Cardiac neuromodulation with acute intermittent hypoxia in rats with spinal cord injury", ID: JP-RP-2025-287676R1

Ahmadian et al

EDITOR COMMENTS

Reviewing Editor:

The manuscript and revisions were reviewed by two reviewers and the reviewing editor. The authors have been responsive to the comments, however, some questions remain from reviewer 2. Specifically, reviewer 2 comments on the lack of explanation on the difference of baselines, the sex difference, and the weak citations concerning translational aspects.

Author's Response: Thank you. We tried to address additional concerns raised by reviewer #2 in the revised manuscript.

Senior Editor:

Thank you for submission of your revised research article to the Journal of Physiology for consideration. Referee 2 has a few additional concerns raised which need to be addressed. Please address all comments from the external referee as well as addressing the list of requirements or publication in the journal.

Please also note that the Data Availability statement requires revision. Data in the manuscript is summary data only. The journal requires authors to permit access to all data incurred in the study if requested in a reasonable fashion and as allowed by any restrictions. (see the Journal's Rigour and Reproducibility document).

We suggest the following wording: The data that support the findings of this study are available from the corresponding author upon reasonable request.

<https://physoc.onlinelibrary.wiley.com/pb-assets/hub-assets/physoc/documents/TJP-Rigour-and-Reproducibility-Requirements-1724673661727.pdf>

Authors' Response: We thank the editor for the opportunity to reconsider our manuscript following a satisfactory revision. We have tried to fully address additional concerns raised by reviewer #2 in the revised manuscript. We have also used your suggested wording for Data Availability statement.

REFEREE COMMENTS

Referee #1:

Thank you for carefully addressing the previous points raised and for making changes to the text. All issues have been satisfactorily addressed.

Authors' response: Thank you!

Referee #2:

Thank you for resubmitting your work to The Journal of Physiology. Below I have provided commentary on specific components addressed by the authors. I hope you find the review helpful, and thank you for your hard work.

Abstract:

No comments

Introduction:

We agree that there is an opportunity for single exposure of IH to be therapeutic, but that is not cohesive with the project. The new terminology helps streamline the paper for clarity.

Stylistically I still think the introduction does not focus in on the relevance of the protocol used in this study, but this is a point I can concede considering the other paper from the same laboratory is now published and presented. No further concerns

Methods:

Results:

Unsure if you want to discuss this in the results or discussion, but I still contend the variability in baseline function should be directly addressed. I do not think it takes away from the paper, but it is difficult to interpret a paper fully when statistical differences are found, but not addressed. This by no means requires extensive discussion, but I do believe it should be addressed.

Authors' response: Thank you for this comment. We have added a statement to mention that the baseline between-group differences were noted for a few metrics in the results section, which reads as below:

“Note that baseline between-group (i.e., TC vs. AIH) differences were noted for a few metrics (i.e., P_{max} , SBP, PP, HR).”

We have additionally added the following section to our discussion:

“It should also be noted that we did find some between-group variability in AIH and TC groups, which manifested as our AIH group having significantly lower values for a number of cardiac indices at baseline. It is likely that this group difference was caused by either subtle differences in the injury characteristics (force/displacement/position) from our contusion injury apparatus, or more likely, inherent within animal differences in the progression of the secondary injury. For example, if animals have a ‘worse’ secondary injury response (i.e., more inflammation and/or prolonged ischemia) then it is expected they will have a more severe injury and present with reduced cardiovascular function. The magnitude of variation across our experimental groups is similar to that previously observed in humans with cervical SCI who are all classified as having a neurologically complete (AIS A) injury”.

Discussion:

Line 523: Ott and Edmunds are investigating sex differences which is not exactly the sexes in this investigation. This should be addressed in the limitations. Likewise, despite the title, Lusina did not complete intermittent hypoxia. The protocol used in that paper was continuous hypoxia over several days, which the differences from day to day is what they deemed "intermittent" (a brief discussion on issues with terminology is available PMID: 37560767 if clarification is needed on this statement). This paper showed increases in MSNA without increases in BP. Thus, I am unsure this is the correct citation for human work. In addition to Ott and Edmunds, acute exposure and alterations in BP in humans could include (ones not cited, if you cited them I did not include them):

PMIDs:

1) LTF of SBP: 35721527

2) LTF of MAP or DBP: 28082332, 31805605

3) LTF of MSNA (albeit they used voluntary apneas with hypoxia: 15242836, 14555683

4) LTF of HRV: 28119623, 18403450

5) Inconsistencies of BP post AIH because of sex: 32966122

6) An in-depth review on human evidence, both within and immediately following AIH is available and may also be suitable for citation. This review covers blood pressure as well as HR: 33781731

Please note these are not required to be cited, but I would consider them better citations to add instead of Lusina. Likewise, these would facilitate any discussion of sex differences for human data. I'm sure there may be a few newer papers, but I am less familiar. I know Univ Florida did publish a manuscript on CV outcomes in SCI (albeit a lot of missing data) that showed no LTF in breathing or BP, that may be worth discussing.

The new text is undercited. This all cannot be a thesis statement. Repeated exposure in those with hypertension, albeit not SCI but still autonomic dysfunction, supports the reduction in BP as well as the BP response (peak) to hypoxia in humans. PMID: 35015980, 35721527 and those after MI: 15262041

Likewise, there are many editorials addressing doses/numbers etc that could be cited here.

I believe the impact, or potential impact, of sex still needs to be addressed considering the translatable papers cited and your rat population. This does not have to be extensive, but it is important to address.

Author's response: Thank you for taking the time to make these comments and point out the additional papers. We have considered the citations you suggested and also addressed some discussion around sex effects. We have also added a brief discussion for a recent study with a focus on cardiorespiratory function in response to AIH treatment post-SCI. Please note that we have decided to stay away from a discussion of how AIH impacts CV function in other disease states that have autonomic dysfunction. The suggested citations, and indeed studies we are aware of, have all addressed whether **DAILY exposure** to AIH alters subsequent CV function. As pointed out in previous comments on the last version, we did not perform daily-AIH. How daily-AIH impacts CV function in SCI rats/humans remains to be examined and thus we believe it does not warrant discussion here.

We have added the following sections of text to the discussion in the relevant sections:

“Such AIH-induced augmentation in sympathetic nerve activity/blood pressure is also reported in able-bodied individuals (Jouett et al., 2017; Ott et al., 2020; Edmunds et al., 2021) and animals with intact spinal cords (Dick et al., 2007; Blackburn et al., 2018; Ostrowski et al., 2023).”

“To our knowledge, only a single recent study has reported on the blood pressure and heart rate response to AIH in individuals with SCI (Welch et al., 2024). The authors reported no significant differences in blood pressure and a small increase in heart rate following AIH exposure. While this work lacks any assessments of cardiac pressure generation capacity, stroke volume or systemic vascular resistance, the lack of increase in blood pressure post-AIH could be due to the more heterogeneous sample used in this clinical work (i.e., inclusion of individuals with injuries ranging from C3 to T6 and AIS grades A-D).”

“Furthermore, our sample included only male rats. Whilst our focus on male rats reflects the much larger proportion of males than females living with chronic SCI (National Spinal Cord Injury Statistical Center, 2023), previous studies have shown sex differences in AIH-induced cardiovascular plasticity (Puri

et al., 2021). Indeed, despite showing similar adjustments in muscle sympathetic activity post-AIH, only males had blood pressure above baseline post-AIH exposure (Jacob et al., 2020). Whether such sex-differences in the blood pressure response to AIH are also present in the setting of SCI remains to be explored”.

Tables and Figures:

Overall Impression:

Overall, I believe the author's addressed most of my concerns adequately, but there some of the new components are undercited considering there are publications that directly address these concerns. The citations for translatable work are somewhat weak and I believe there are better citations- some examples were provided. Please note, none of this, in my opinion, requires extensive citations or discussion, but an easy opportunity to improve precision of the translatability of your work to humans.

Authors' response: Thank you for your additional comments and time spent reviewing our work. We have tried to address these additional concerns in the revised manuscript.

Referee #3:

Thank you to the authors for responding to my comments. I believe they have adequately addressed my concerns.

Authors' response: Thank you!!

Dear Dr West,

Re: JP-RP-2025-287676R2 "Cardiac neuromodulation with acute intermittent hypoxia in rats with spinal cord injury" by Mehdi Ahmadian, Erin Erskine, and Christopher R West

We are pleased to tell you that your paper has been accepted for publication in The Journal of Physiology.

Yours sincerely,

Harold Schultz
Senior Editor
The Journal of Physiology

If you would like to receive our 'Research Roundup', a monthly newsletter highlighting the cutting-edge research published in The Physiological Society's family of journals (The Journal of Physiology, Experimental Physiology, Physiological Reports, The Journal of Nutritional Physiology and The Journal of Precision Medicine: Health and Disease), please click this link, fill in your name and email address and select 'Research Roundup':
<https://www.physoc.org/journals-and-media/membernews>

- **TRANSPARENT PEER REVIEW POLICY:** To improve the transparency of its peer review process, The Journal of Physiology publishes online as supporting information the peer review history of all articles accepted for publication. Readers will have access to decision letters, including Editors' comments and referee reports, for each version of the manuscript as well as any author responses to peer review comments. Referees can decide whether or not they wish to be named on the peer review history document.
- You can help your research get the attention it deserves! Check out Wiley's free Promotion Guide for best-practice recommendations for promoting your work at: www.wileyauthors.com/eeo/guide. You can learn more about Wiley Editing Services which offers professional video, design, and writing services to create shareable video abstracts, infographics, conference posters, lay summaries, and research news stories for your research at: www.wileyauthors.com/eeo/promotion.
- **IMPORTANT NOTICE ABOUT OPEN ACCESS:** To assist authors whose funding agencies mandate public access to published research findings sooner than 12 months after publication, The Journal of Physiology allows authors to pay an Open Access (OA) fee to have their papers made freely available immediately on publication.

EDITOR COMMENTS

Reviewing Editor:

The authors have adequately addressed all the reviewers and editors' concerns.

Senior Editor:

The editors wish to thank the authors for these final adjustments to the manuscript. The article is now accepted for publication. Congratulations for an interesting and insightful study. Please consider the Journal of Physiology for your future studies.

REFEREE COMMENTS

Referee #2:

No further comments. The authors have responded with adequate text considering the sections that were asked for to be improved.

However, the authors should note, despite several citations focusing on daily exposure to IH, most human studies still report the first and last day of hypoxic exposure. Thus, data on acute exposure is available. A significant portion of the review from Puri et al (cited) used such information for the review.